# Photo-excited extracellular electron transfer of electroactive microorganism triggers RAFT polymerization

Chao Li[1,2,3,10], Jing Liu[4,10], Wenchang Hu[2,3], Lin Xiao[2,3], Feng Li[2,3], Qijing Liu[2,3], Junqi Zhang[1], Huan Yu [1], Baocai Zhang[1], Dake Xu [5,6], Shaoan Cheng [7], Wen-Wei Li [8], Kenneth H. Nealson[9] & Hao Song [1,2,3] ✉

Living cell-triggered reversible addition-fragmentation chain-transfer (RAFT) polymerization is of great value for construction of living materials with diverse applications. However, microorganisms-activated polymerization without end-group heterogeneity is not yet established. Here, we develop an electroactive microorganism-triggered polymerization system using *Shewanella oneidensis*-secreted flavins (as electron shuttles) to directly reduce chain transfer agents (CTAs) to continuously generate radicals, thus initiating RAFT polymerization. This *S. oneidensis*-triggered polymerization integrates microbial extracellular electron transfer pathway and photoinduced electron transfer to reduce CTAs for continuous radical generation. We then genetically engineer *S. oneidensis* to enhance flavins biosynthesis and transport, accomplishing increased conversion ratio ( > 90%) of poly(N, N-dimethylacrylamide) with low polydispersity ($Đ < 1.20$). In addition, the *S. oneidensis*-triggered RAFT polymerization is effective for various monomers and CTAs, being able to synthesize diverse block copolymers. Synergistic integration of synthetic biology and RAFT polymerization provides a sustainable and controllable polymerization platform.

Reversible deactivation radical polymerization (RDRP) technologies, including atom transfer radical polymerization (ATRP) and reversible addition-fragmentation chain-transfer (RAFT) polymerization, are powerful tools for synthesizing well-defined polymers with controlled molecular weight and its distributions, chemical composition, and chain topology and architecture[1–4]. Since these techniques enable appropriate controllability of polymers with various monomers, they have been adopted in numerous applications, including surface and interface engineering[5,6], 3D printing[7,8], polymer networks

preparation[9–11], molecular imprinting[12,13], and polymer-biomolecule bioconjugation[14–16]. Living cell-mediated RDRP is of great value for the construction of engineered living materials, since it combines biotechnology and polymer chemistry to eco-friendly produce precisely controlled polymers under mild conditions[17–20]. In particular, due to excellent biocompatibility of the radical polymerization processes, the application of RDRP for the modification of living cells has attracted widespread attention[17], e.g., extracellularly synthesized polymers via RDRP were used for cellular encapsulation or engineering cell

[1]College of Life and Health Sciences, Northeastern University, Shenyang, China. [2]State Key Laboratory of Synthetic Biology, Tianjin University, Tianjin, China. [3]School of Synthetic Biology and Biomanufacturing, Tianjin University, Tianjin, China. [4]Institute of Entomology, College of Life Sciences, Nankai University, Tianjin, China. [5]Shenyang National Laboratory for Materials Science, Northeastern University, Shenyang, China. [6]Electrobiomaterials Institute, Key Laboratory for Anisotropy and Texture of Materials (Ministry of Education), Northeastern University, Shenyang, China. [7]State Key Laboratory of Clean Energy, Department of Energy Engineering, Zhejiang University, Hangzhou, China. [8]Chinese Academy of Sciences Key Laboratory of Urban Pollutant Conversion, Department of Environmental Science and Engineering, University of Science & Technology of China, Hefei, China. [9]Departments of Earth Science & Biological Sciences, University of Southern California, South Pasadena, CA, USA. [10]These authors contributed equally: Chao Li, Jing Liu. ✉e-mail: songhao@mail.neu.edu.cn

surfaces[21–23], and intracellularly synthesized polymers were employed to modulate the function and behavior of living organisms and create biohybrids or engineered living materials with adaptive, programmable capabilities to facilitate cellular therapy[17,19,22,24–29]. In most of these cases, polymerization occurred in cells is triggered by external stimuli (e.g., light), while metabolism of the cells has negligible impact or regulation on the polymerization processes. In this regard, it is of utmost importance to construct microorganisms triggered in-situ RDRP polymerization systems that harness metabolic pathways to initiate and produce synthetic polymers, which would promote the development of synthetic materials with living functions[18,30–34].

A number of living cells-activated ATRP processes were achieved by employing the metal reducing activity of bacteria, which could mediate the active and dormant states of the copper, iron, or other metallic catalysts, thus activating the alkyl bromide initiators to enable ATRP, as demonstrated by the pioneering works of the Alexander, Keitz and Rawson groups[20,35–39]. On the other hand, RAFT polymerization employed chain-transfer agents (CTAs, i.e., the thiocarbonylthio compounds) to achieve an equilibrium between propagating and dormant species via a degenerative chain-transfer mechanism[1,40]. RAFT polymerization essentially required continuous supply of radicals to initiate and sustain polymerization[1], which posed challenges for the development of living organisms-activated RAFT. The Qiao group pioneered the living microorganism-activated RAFT polymerization, demonstrating that the bacterial terminal electron pathway can be harnessed to reduce the exogenous aryl diazonium initiators, which generated aryl radical species to initiate RAFT polymerization[41]. Although living organisms can activate exogenous radical initiators to produce radicals for RAFT polymerization, the use of exogenous radical initiators led to end-group heterogeneity and chain termination, which was an inherent issue of using free radical initiators in RAFT polymerization[42–45]. Radicals generated from exogenous initiators could not only cause continuous generation of new chains (initiator-derived polymer chains) to reduce the molecular weight of target polymers, but also lead to gradual accumulation of terminated chains to increase the dispersity of polymers[46,47]. These side effects could be further amplified in the synthesis of block copolymers, hampering the preparation of pure block polymers[48,49]. In addition, in order to achieve high polymerization conversion ratio in RAFT polymerization, a high concentration of radical initiators was generally required, which would lead to serious cytotoxicity and restrict development of living cells-activated RAFT polymerization systems[41,50]. To eliminate the impact of initiators on the quality of the synthesized polymers, a number of external stimulus methods were recently developed to regulate RAFT polymerization, such as light or electricity, which could trigger electron transfer to reduce the CTA to form radicals that subsequently initiated RAFT polymerization[46,51]. Exploring bio-reduction of CTAs as the radical initiation system is thus expected as an alternative strategy for the construction of living cells-activated RAFT polymerization processes. Nevertheless, under physiological conditions, the reduction potential of the electron shuttles produced by microorganisms was generally higher than that of CTAs[52–55], which made the electron shuttles thermodynamically infeasible to reduce CTAs (i.e., the thiocarbonylthio compounds) that generally had lower reduction potentials[56,57]. Consequently, this issue posed challenge to the development of living cells-activated in situ RAFT polymerization.

In this study, we developed an electroactive microorganism-triggered RAFT polymerization system using *Shewanella oneidensis*-secreted electron shuttles, i.e., flavins (including riboflavin and flavin mononucleotide (FMN)) as the electron mediators to reduce CTAs to continuously generate radicals, which then initiated RAFT polymerization via the combined processes of bioreduction and photo-excitation. As illustrated in Fig. 1, this *S. oneidensis*-triggered RAFT polymerization system was comprised of three components: 1) flavins as the electron mediators were electrically reduced to its fully reduced

form of flavins hydroquinone ($FL_{hq}$) by the endogenous EET pathway of *S. oneidensis*, which eliminated the need of exogenous addition of radical initiators in the RAFT polymerization process; 2) $FL_{hq}$ as the electron mediators to generate the excited form $FL_{hq}^*$ under photo-irradiation, displaying high reduction potential that could reduce CTAs via photoexcited electron transfer, thus, overcoming the thermodynamic hurdle of the electron transfer from $FL_{hq}$ to CTAs; and 3) an genetically engineered *S. oneidensis* strain with enhanced biosynthesis and transport ability of flavins to increase the monomer conversion ratio of RAFT polymerization, which also avoided the need of exogenous addition of flavins.

Firstly, we investigated the electrical and photophysical properties of flavins, demonstrating that $FL_{hq}$, the reduced form of flavins generated by the extracellular electron transfer (EET) of *S. oneidensis*, can overcome the thermodynamic barrier of electron transfer from $FL_{hq}$ to CTAs upon photoexcitation. Therefore, flavins, as electron mediators, can effectively shuttle intracellular electrons from microbial cells to CTAs, thereby initiating RAFT polymerization under light irradiation. Secondly, we used synthetic biology approaches to heterologously incorporate the flavins biosynthesis pathway from *Bacillus subtilis* and the pore protein OprF from *Pseudomonas aeruginosa* into *S. oneidensis*, which enabled endogenous secretion of flavins at ~21 μM. The engineered *S. oneidensis* strain with increased flavins concentration could thus significantly promote the RAFT polymerization reaction rate and synthesize predictable polymer products with consistently low dispersity ($Đ < 1.20$) and high monomer conversion ratio ($> 90\%$). Thirdly, we showed that the *S. oneidensis*-triggered RAFT polymerization system was effective for a variety of monomers and CTAs as well as for the preparation of well-defined block copolymers. Overall, this constructed *S. oneidensis*-triggered RAFT polymerization system utilizes D-lactate as the electron source via cellular metabolism, and employs endogenously synthesized flavins as electron mediators to reduce CTAs under light irradiation. This enables continuous radical generation, exhibiting high electron transfer efficiency from the *S. oneidensis* cells to the extracellular electron acceptors (thiocarbonylthio-based CTAs), thereby initiating and sustaining the RAFT polymerization. This system is a proof-of-concept demonstration of achieving living microorganism-triggered RAFT polymerization without the need of exogenous radical initiators, avoiding end-group heterogeneity and chain termination of the resulting polymers.

## Results and discussion
### Construction of flavins-mediated RAFT polymerization system with photoexcited extracellular electron transfer of *S. oneidensis*

In traditional RAFT polymerization, exogenous radical initiators may cause cytotoxicity to cells and lead to end-group heterogeneity in polymer products. Thus, to eliminate the need for exogenous addition of radical initiators, we constructed an electroactive microorganism-triggered RAFT polymerization system, in which *S. oneidensis*, an extensively studied model exoelectrogen[58–61], metabolizes D-lactate (serving as both the carbon source and the electron donor) to continuously generate electrons to trigger radical polymerization. In this system, flavins (riboflavin and FMN) synthesized by *S. oneidensis* act as the electron mediators. Flavins synthesized and secreted from *S. oneidensis* were proven to play a key role as electron shuttles to mediate EET[58]. We thus examined whether flavins as the redox mediators could shuttle electrons from cells to the thiocarbonylthio CTAs to initiate the radical polymerization (Fig. 2a). We tried to conduct the radical polymerization of N, N-dimethylacrylamide (DMA) by using exogenously added flavins (oxidized state) as the redox mediator, 2-ethylsulfanylthiocarbonylsulfanyl-propionic acid methyl ester as the CTA (denoted as CTA1), and the wild-type *S. oneidensis* MR-1 as the living electrode under anaerobic condition.

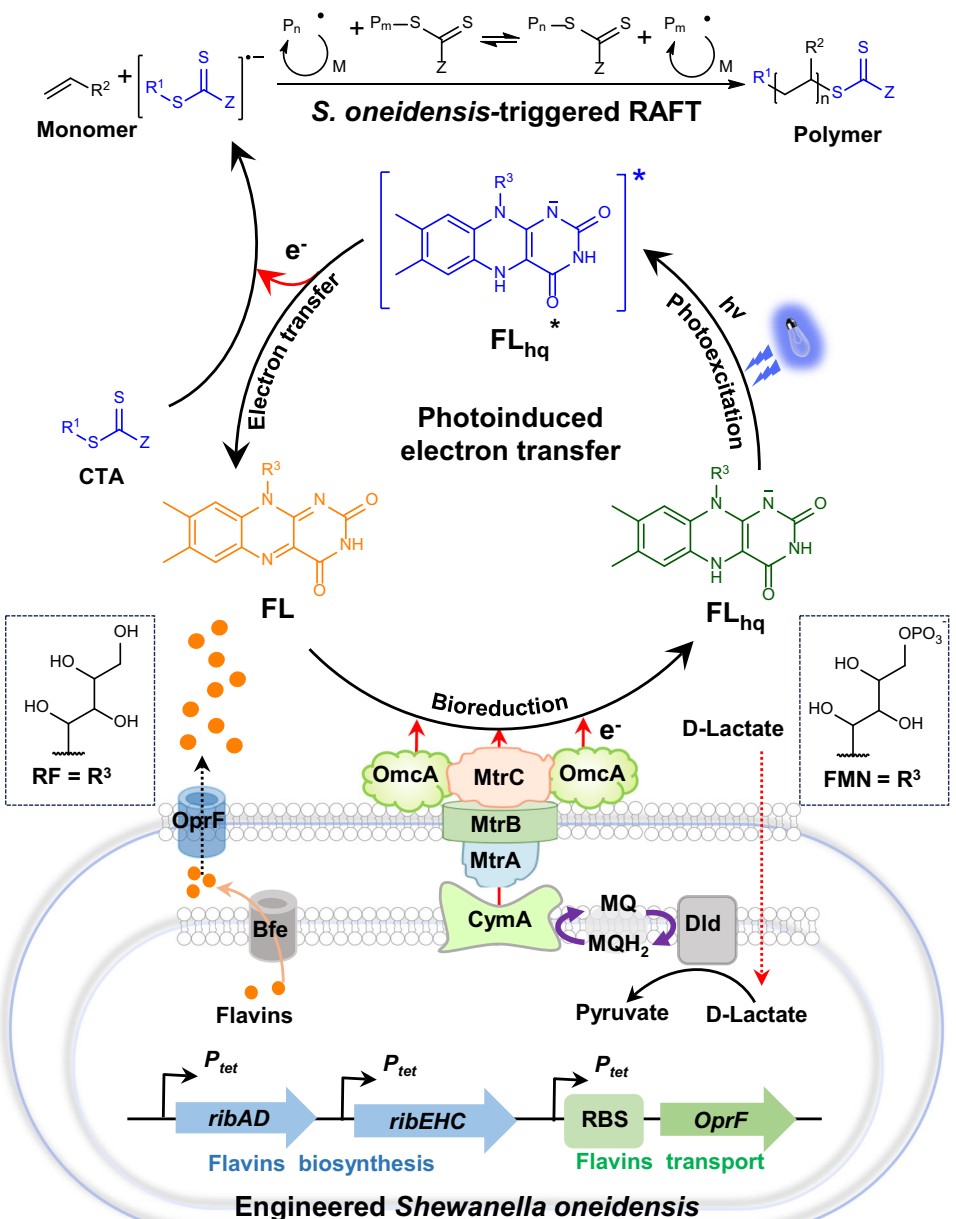

**Fig. 1 | Construction of the electroactive microorganism-triggered reversible addition-fragmentation chain-transfer (RAFT) polymerization system by photoexcited extracellular electron transfer of the genetically engineered *Shewanella oneidensis*.** First, the engineered *S. oneidensis* was constructed by synthetic biology approach to heterologously express the flavins synthesis pathway from *Bacillus subtilis* and the pore protein OprF from *Pseudomonas aeruginosa* in *S. oneidensis*, which enhanced flavins biosynthesis and transportation. Second, flavins acted as the electron mediators that integrated microbial extracellular electron transfer (EET) and photoinduced electron transfer (PET) to reduce the chain transfer agent (CTA, i.e., thiocarbonylthio) for generation of free radicals, which triggered the microorganism-mediated RAFT polymerization. In this process, the electron transfer pathways included three steps: (i) production of electrons by engineered *S. oneidensis* through D-lactate metabolism via intracellular electron transfer process; (ii) flavins reduction via the EET process; and (iii) CTA reduction via PET. Specifically, the electrons produced by metabolizing D-lactate in *S. oneidensis* transferred via the cytoplasmic membrane protein CymA, which reduced periplasmic proteins that transferred electrons to the Mtr-pathway (i.e., electrons transferred through MtrA to MtrC and OmcA). Then, electrons transferred to the endogenously secreted flavins. Bioreduction of flavins (FL) via EET generated the fully reduced flavins hydroquinone ($FL_{hq}$). Photoexcitation of $FL_{hq}$ led to the formation of a photoexcited intermediate $FL_{hq}^*$, which reduced CTA via PET to produce free radicals, initiating the RAFT polymerization.

However, the polymerization reaction could not occur using either riboflavin or FMN as the redox mediator under this condition (Fig. 2b). We firstly examined the electron transfer process from the cells to flavins using the UV-vis absorption spectroscopy, which suggested that the yellow-colored riboflavin or FMN was reduced to its fully reduced form of flavin hydroquinone ($FL_{hq}$) in the culture media of *S. oneidensis*[62] (Supplementary Fig. 1). We then analyzed the electrochemical properties of flavins and a series of CTAs by cyclic voltammetry (CV) (Supplementary Figs. 2 and 3). The reduction potential of the fully reduced flavins hydroquinone ($FL_{hq}$) ($E_{red}$ ($Riboflavin_{hq}$) = −0.45 V vs. Ag/AgCl; and $E_{red}$ ($FMN_{hq}$) = −0.49 V vs. Ag/AgCl)[63,64] was more positive than that of CTAs ($E_{red}$ (CTA) = −0.7 V - −0.99 V vs. Ag/AgCl), and the calculated Gibbs free energy of electron transfer ($\Delta G_{EET}$) from $FL_{hq}$ to CTAs was positive ($\Delta G_{EET} > 0$) (Fig. 2c, and Supplementary Table 1), suggesting the reduction reaction of CTAs by flavins was thermodynamically infeasible, thus initiating radicals could not be generated under this condition.

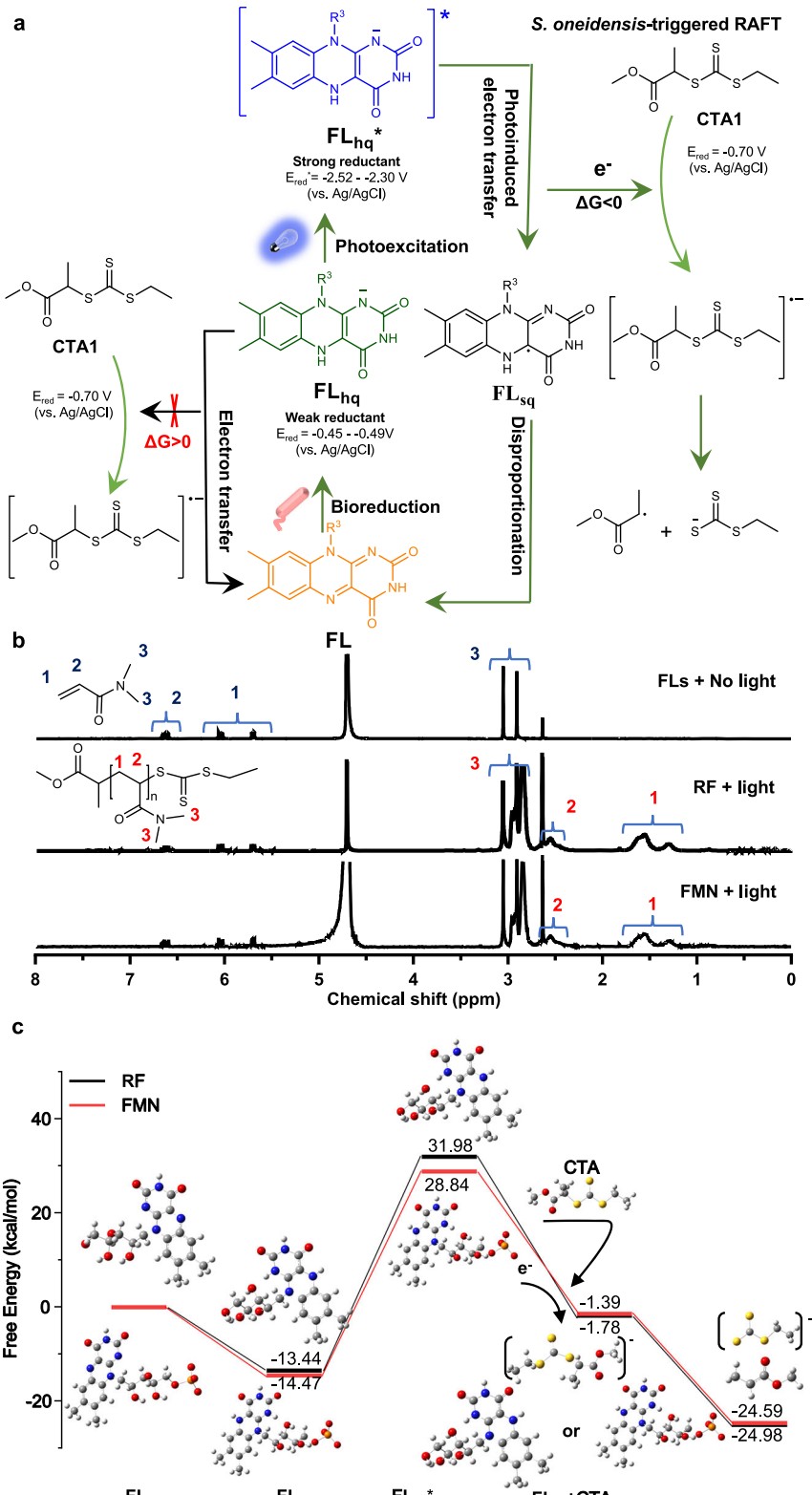

**Fig. 2 | Insights from thermodynamic analyses of the *S. oneidensis*-triggered RAFT polymerization. a** Electrons generated from *S. oneidensis* reduce flavins from the oxidized flavins to the fully reduced $FL_{hq}$. The reduced electron mediator $FL_{hq}$ reacts with CTA1 (2-ethylsulfanylthiocarbonylsulfanyl-propionic acid methyl ester) to produce radical (black arrow, left), whereas the reduction reaction was thermodynamically infeasible. Upon the blue light irradiation, $FL_{hq}$ was excited to its excited state $FL_{hq}^*$, which enabled transfer of the electrons to CTA1 to generate radicals via PET (green arrow, right). **b** $^1$H NMR spectra of the monomer N,

N-dimethylacrylamide (DMA) in the mixture of *S. oneidensis*, DMA, CTA1, and exogenously added flavins (riboflavin (RF) or FMN), where no radical polymerization was observed. And $^1$H NMR spectra of the Poly(N, N-dimethylacrylamide) (PDMA) polymer synthesized in the polymerization system upon photoexcitation using riboflavin or FMN as the electron mediator, respectively. **c** Calculated Gibbs energy profile of the electron transfer pathways from flavins (riboflavin or FMN) to CTA1. Source data were provided as a Source Data file.

Photoinduced electron transfer (PET) is a widely utilized mechanism that enables electron transfer events that are otherwise unattainable for molecules in their ground state[65,66]. It was recently found that flavoenzymes can utilize highly reducing excited flavin hydroquinone to participate in PET reactions, thereby initiating radical reactions within the enzyme active site[63,67–69]. Inspired by these findings, to overcome the thermodynamic barrier of electron transfer, we hypothesized that $FL_{hq}$ maybe act as a photocatalyst capable of reducing the CTA through the PET process, which may lead to the initiation of the radical polymerization reaction. Thus, we used blue light to irradiate the culture media of *S. oneidensis* MR-1 with DMA (as the monomer), CTA1 and flavins, which resulting in radical polymerization to form PDMA with high conversion ratio (> 90%) of DMA, as detected by the $^1H$ NMR spectra (Fig. 2b). To elucidate the reaction mechanism, density functional theory (DFT) was employed to calculate the excited-state reduction potentials of flavins hydroquinone ($FL_{hq}^*$) ($E_{red}$ (Riboflavin$_{hq}^*$) = −2.52 V vs. Ag/AgCl, and $E_{red}$ (FMN$_{hq}^*$) = −2.30 V vs. Ag/AgCl[63]), which were more negative than that of the CTAs (−0.7 V - −0.99 V vs. Ag/AgCl). This difference in reduction potentials provided sufficient driving force to enable electron transfer from the excited state of $FL_{hq}^*$ to CTA (Supplementary Fig. 4, Supplementary Tables 2–4, and Supplementary Data 1; for details, see Supplementary Information). In addition, the Gibbs free energy $\Delta G_{PET}$ of the PET from the excited $FL_{hq}^*$ to CTA1 were calculated to be −33.76 kcal mol$^{-1}$ (from Riboflavin$_{hq}^*$) and −30.23 kcal mol$^{-1}$ (from FMN$_{hq}^*$), respectively (Fig. 2c), which supported the hypothesis that the reduction of CTAs by $FL_{hq}^*$ was thermodynamically feasible. Thus, upon accepting electrons from microbial metabolism of lactate, flavins are then photoexcited and subsequently reduce CTA via PET.

Then, a set of control experiments were conducted, revealing that the necessary components, including monomer (DMA), CTA, *S. oneidensis*, flavins and light, were required for the RAFT polymerization to occur (Supplementary Table 5, and Supplementary Figs. 5–8). In the absence of CTA1, the polymerization was uncontrollable ($Đ = 2.6$), since the reduction potential of $FL_{hq}^*$ was more negative than that of DMA, thus the reduction of DMA to generate radicals could initiate polymerization of the monomers. Especially, no polymerization was detected when using the free oxidation state of flavins in lieu of *S. oneidensis*-reduced flavins, which was consistent with the previously reported literatures[70,71]. This observation convincingly evidenced that flavins, as the electron mediator, gained its photocatalytic capability only after reduction to $FL_{hq}$ by *S. oneidensis*. When $FL_{hq}$ was supplied alone, the monomer conversion ratio was rather low (< 25%), indicating that $FL_{hq}$ was consumed in the course of PET and could not be regenerated to sustain polymerization. This observation demonstrated that the continuous secretion of extracellular electrons by the viable cells to regenerate $FL_{hq}$ is a crucial step. Furthermore, we conducted time-staged cell-killing experiments at different time points (0.5 h, 1 h, and 2 h) in the polymerization reaction. When the cells were killed in the course of polymerization process, the reaction stopped immediately, confirming that live cells were essential not only at the initiation stage but also throughout the entire polymerization process (Supplementary Fig. 9). We then evaluated the cell viability of *S. oneidensis* under optimized polymerization conditions. With riboflavin or FMN serving as the electron mediator, the cell viability determined by the colony-forming units (CFU) assay showed a slow decline to 89% and 90% of its original cell viability after 2 hours' polymerization, and to 81% and 80% after 6 hours' polymerization, during which the monomer conversion ratios exceeded 90% (Supplementary Fig. 10). Thus, this RAFT polymerization reaction system was essentially biocompatible with *S. oneidensis*. Taken together, these results supported the successful construction of the *S. oneidensis*-triggered polymerization system that was capable of initiating and sustaining radical polymerization.

To validate the *S. oneidensis*-triggered radical initiation system for the radical polymerization, we investigated the electron transfer process of EET and PET from the cells to CTA. Firstly, we demonstrated the EET process from microbial cells to flavins using UV-vis absorption spectroscopy. Flavins could be reduced to its fully reduced form $FL_{hq}$ via the MtrCAB-based EET pathway of *S. oneidensis* (as shown in the bioreduction step in Figs. 1, and 2a), in which the electrons were generated from the cell metabolism of D-lactate (the carbon source). Secondly, to prove that the PET between $FL_{hq}$ and CTA could indeed occur, fluorescence quenching experiments were performed. With the addition of CTA, significant change in the fluorescence intensity proved the electron transfer between $FL_{hq}$ and CTA (Supplementary Figs. 12a and 14a). However, only a slight change in fluorescence intensity was observed with the addition of DMA (Supplementary Figs. 12b and 14b), suggesting that the electrons preferentially transfer from flavins to CTA, thereby enabling controlled polymerization. Additionally, due to the very weak fluorescence emissions of $FL_{hq}$ in the solution, time-resolved fluorescence spectroscopy was further used to analyze the electron transfer in the dynamic quenching process. When oxidation state flavins were reduced by sodium dithionite ($Na_2S_2O_4$, as the electron donor) and excited at 405 nm, the fluorescence lifetimes of $RF_{hq}$ and $FMN_{hq}$ were 3.52 ns and 3.38 ns, respectively (Supplementary Figs. 15 and 16). In the presence of CTA1, the fluorescence lifetimes of $RF_{hq}$ and $FMN_{hq}$ decreased to 3.01 ns and 2.23 ns, strongly indicating CTA1 was reduced by $FL_{hq}$ in the PET process. Moreover, the electron spin resonance (ESR) experiment was conducted to confirm the radical formation. 5,5-dimethyl-1-pyrroline N-oxide (DMPO) was mixed with riboflavin, $Na_2S_2O_4$ and DMA in $H_2O$ (with a molar ratio of RF: $Na_2S_2O_4$: DMA: DMPO = 1: 10: 100: 10). In the blue irradiation experiment, a weak six-line ESR signal emerged with hyperfine splitting constants of $\alpha^N = 15.2$ G and $\alpha^H = 22.9$ G, which represented a characteristic ESR pattern of the DMPO-alkyl radical adduct and revealed radical generation during the polymerization (Supplementary Fig. 17)[25,70].

Thus, in this constructed *S. oneidensis*-triggered RAFT polymerization system, flavins accepted electrons from *S. oneidensis* and was converted to the reduced $FL_{hq}$ via microbial EET, followed by photoexcitation to form the excited $FL_{hq}^*$, which had sufficiently low electrical potential to reduce the CTA via PET to form radicals, initiating radical polymerization. $FL_{hq}$ lost one electron to form flavins semiquinone ($FL_{sq}$) through oxidative quenching, which was consistent with the predominant involvement of single-electron processes in the photocatalysis of flavins[62,63,69]. However, due to the inherent instability of $FL_{sq}$ in the solution[62,64,72], we proposed that the majority of $FL_{sq}$ reverted to the oxidized state of flavins through a disproportionation reaction, thereby completing the catalytic cycle. The formation of the oxidized state of flavins was further confirmed through fluorescence quenching experiments. As shown in Supplementary Figs. 12 and 14, along with the increase in the CTA concentration, the fluorescence intensity of the solution also increased, which was attributed to the strong fluorescence emission of the oxidized state of flavins (Supplementary Figs. 11 and 13).

## Controlled radical polymerization enabled by flavins-mediated polymerization

To examine the controllability of the RAFT polymerization reaction, the polymerization kinetic studies were performed using riboflavin or FMN as the electron mediator, respectively. As shown in Fig. 3a, a linear correlation of ln([M$_0$]/[M]) with the reaction time corresponded to pseudo first-order polymerization kinetics for polymerization reactions, which indicated a constant radical concentration of propagating radicals for this RAFT polymerization system[73,74]. Notably, the two mediators (riboflavin and FMN) exhibited similar polymerization rates (determined by the slope of ln([M$_0$]/[M]) vs. reaction time), indicating that riboflavin and FMN possess comparable catalytic activities. The

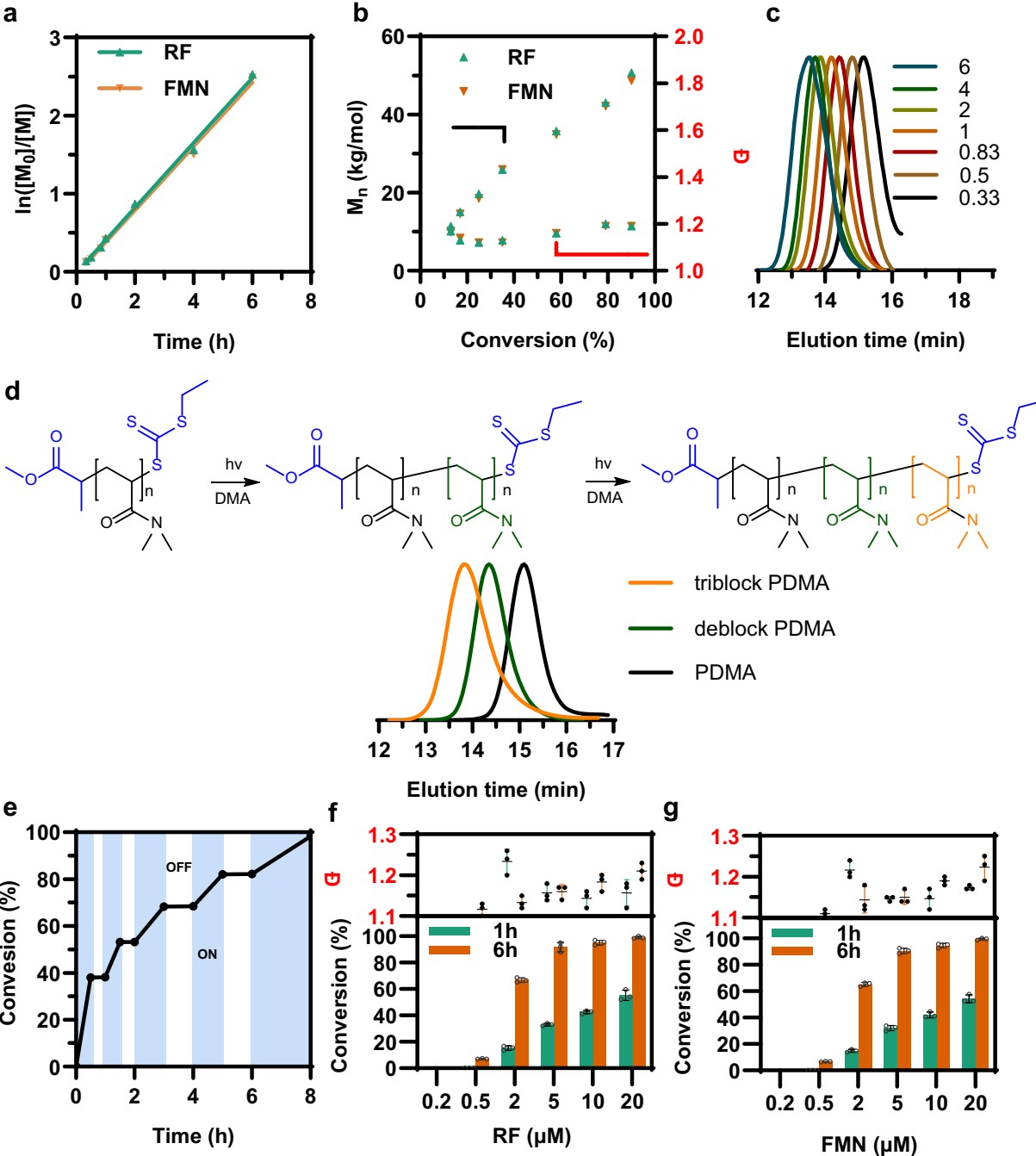

**Fig. 3 | Kinetic results of the *S. oneidensis*-triggered RAFT polymerization with exogenously added flavins (riboflavin and FMN). a** First-order kinetics of the microorganism-triggered RAFT polymerization. **b** Molecular weight (black circles) and dispersity (red squares) evolution. **c** GPC traces of PDMA synthesized at various polymerization time. **d** Gel permeation chromatography (GPC) traces of block copolymer. **e** Temporal control of the *S. oneidensis*-triggered RAFT polymerization with intermittent light. **f** Monomer conversion ratio and polydispersity (Đ) by employing different riboflavin (RF) concentrations upon 1 h and 6 h polymerization reaction. **g** Monomer conversion ratio and polydispersity (Đ) by employing different FMN concentrations upon 1 h and 6 h polymerization reaction. Data showed mean ± SD of three independent experiments. Data in (**f**, **g**) were shown as the mean ± standard deviation (SD) (*n* = 3). Source data were provided as a Source Data file.

molecular weight exhibited a linear increase with the conversion and aligned well with the theoretical value (Fig. 3b). The gel permeation chromatography (GPC) traces indicated a narrow molecular weight distribution (Fig. 3c, and Supplementary Table 6). These kinetic data suggested that the microorganism-triggered polymerization corroborated well with the characteristics of the controlled radical polymerization. Notably, an induction period prior to the polymerization

reaction was not observed as monitored in the previous PET-RAFT studies[50], further highlighting that the system could enhance the free radical initiation process in RAFT polymerization.

To further determine the terminal fidelity, in situ chain extensions of PDMA polymers were performed using DMA as the monomer to yield the block copolymer PDMA-*b*-PDMA. Initially, using riboflavin as the electron mediator, the polymerization of DMA was carried out to

produce a starting PDMA homopolymer ($M_n$ = 10.3 kg mol$^{-1}$, Đ = 1.12) (Supplementary Table 7). Then, the deblock copolymer was synthesized by sequential monomer addition in one pot to produce a PDMA-*b*-PDMA ($M_n$ = 21.7 kg mol$^{-1}$, Đ = 1.15), which mitigated the need for elaborate polymer purification to afford efficient chain extension. Subsequently, we focused on in situ synthesis of triblock copolymers, achieving molecular weight of 29.3 kg mol$^{-1}$ with dispersity of 1.37. GPC traces showed a clear shift of the starting macroinitiators to high molecular weight with narrow distribution (Fig. 3d). These results confirmed that the polymers synthesized via *S. oneidensis*-triggered RAFT polymerization maintained high terminal fidelity[25,50]. It is important to note that this *S. oneidensis*-triggered RAFT polymerization operates by using CTA instead of an initiator, thereby suppressing the end-group heterogeneity of polymers and the termination of the propagating chain[42].

In the *S. oneidensis*-triggered RAFT polymerization, light activation is necessary to invoke the PET from FL$_{hq}$ to CTA. To demonstrate the photo-mediated temporally controlled polymerization, we exposed the polymerization system to an alternating light ON and OFF environment. As shown in Fig. 3e, no polymerization occurred in the absence of light (light OFF). When the light was turned ON, polymerization proceeded as anticipated. GPC analysis of the PMA polymers produced in the course of the light ON/OFF experiment demonstrated good control over polymerization (Supplementary Fig. 18), consistent with the results from the uninterrupted light experiments. The final monomer conversion ratio of 98% was achieved, with a narrow dispersity (Đ = 1.15) (Supplementary Table 8). In addition, to assess the long-term controllability of the polymerization process, we extended the duration of the light ON/OFF intervals (Supplementary Fig. 19). After 24-h reaction, the final monomer conversion reached 91%, confirming the robustness and temporal controllability of this microorganism-triggered RAFT polymerization system. To further assess the viability of *S. oneidensis* over the course of long-term reaction, we performed colony-forming unit (CFU) assays. The CFU results showed that approximately 65% of the *S. oneidensis* cells remained viable, indicating that the polymerization system maintains good biocompatibility, allowing *S. oneidensis* to sustain sufficient viability over extended timeframes to support continuous electron transfer and efficient biocatalytic polymerization.

Radical polymerization usually requires strictly anoxic conditions since the initial radicals could be quenched in the presence of oxygen[75]. *S. oneidensis* is a facultative anaerobe and preferentially respires on oxygen, which has been reported to quickly consume dissolved oxygen by microbial metabolism in ATRP[37]. Therefore, to verify the dissolved oxygen tolerance of our method without degassing the reaction mixtures to retain the active RAFT polymerization, we measured the feasibility of aerobic polymerization. As shown in Supplementary Fig. 20, the Vial-1 (anaerobic condition) achieved increased monomer conversion ratio (37%) compared to the Vial-2 (32% in aerobic condition) within the initial reaction of 1 h. The initial polymerization rate of the system without oxygen removal was slightly lower than that of the de-oxygenated system, due to dissolved oxygen in the mixture quenched radicals to inhibit polymerization. With the elimination of oxygen by the respiration of *S. oneidensis* cells in the polymerization media, the polymerization rate of Vial-2 reached a similar level to that of Vial-1, and both eventually achieved monomer conversion ratio of ~80% after 4 h reaction. These results suggested that the *S. oneidensis*-triggered RAFT system had good oxygen tolerance due to fast consumption and elimination of dissolved oxygen by *S. oneidensis*.

After successfully establishing the flavin-mediated RAFT polymerization system, we studied the impact of the concentration of exogenously added flavins on the RAFT polymerization. As shown in Fig. 3f, g, increasing the flavins concentration from 0.2 to 20 μM, the polymerization reaction rate increased linearly. Remarkably, when the

riboflavin or FMN concentration was above 5 μM, the monomer conversion ratio reached 33% or 32% after 1 hour's polymerization, and 95% or 90% after 6 h polymerization, respectively. It is worth mentioning that an increase in flavins concentration accelerated polymerization rates with minor effect on the polydispersity (Đ < 1.25) and molecular weight control. These results suggested that the concentration of flavins used in the microorganism-triggered RAFT polymerization was low, which may eliminate the need for flavins removal as a post-polymerization process. Since the cell-secreted flavins have been demonstrated to dominate the EET of *S. oneidensis*[58], we further assessed the polymerization activity using the *S. oneidensis* secreted flavins as the electro-photocatalyst to replace the exogenously added flavins. Since *S. oneidensis* secretes a mixture of riboflavin and FMN[58,76], we examined the effect of varying ratios of riboflavin and FMN mixtures on the polymerization rate. As shown in Supplementary Fig. 21, the results showed that the flavin mixtures with different ratios enabled similar polymerization rates, further suggesting that riboflavin and FMN possess comparable catalytic activities. However, the concentration of endogenously synthesized flavins by the wild-type *S. oneidensis* MR-1 was low (~0.54 μM), which resulted in low monomer conversion ratio (<17%). Although exogenous addition of commercially available riboflavin or FMN in the RAFT polymerization reaction could significantly elevate the conversion, the cost of riboflavin or FMN as an expensive additive would restrict their use in large-scale polymerization applications. Therefore, it is crucial to develop an engineered *S. oneidensis* strain with enhanced endogenous flavins production to control the RAFT polymerization performance.

## Enhanced flavins biosynthesis and transport to accelerate monomer conversion ratio of *S. oneidensis*-triggered RAFT polymerization

To increase the flavins level in the *S. oneidensis*-triggered RAFT polymerization system, we developed a modular synthetic biology strategy to improve the de novo biosynthesis and transport of flavins. Accordingly, the biosynthesis pathway of flavins from *Bacillus subtilis* was heterologously expressed in *S. oneidensis* (Fig. 4a). Several promoters compatible with *S. oneidensis* were systematically studied to strengthen the expression of the flavin biosynthesis pathway and boost flavins production. The synthetic flavins biosynthesis gene cluster *ribADEHC* was constructed under the regulation of multiple inducible promoters, including ArcA, Bad, Tac, and Tet, which resulted in the construction of recombinant *S. oneidensis* strains named P$_{arcA}$, P$_{bad}$, P$_{tac}$, P$_{tet}$, respectively (Supplementary Table 9). As shown in Fig. 4b, the recombinant *S. oneidensis* strains P$_{bad}$, P$_{tac}$, and P$_{tet}$ could produce higher concentration of flavins than the recombinant strains of the constitutive promoter P$_{arcA}$ and the wild-type *S. oneidensis* MR-1. In particular, the *S. oneidensis* strain P$_{tet}$ could produce 17.6 μM flavins, which was 1.5-fold higher than that of the strain P$_{bad}$ (4.1 μM) and 4.3-fold higher than that of the strain P$_{tac}$ (11.8 μM) (Fig. 4b, and Supplementary Fig. 22). Since the polymerization rate depended linearly on the flavins concentration, the engineered strain P$_{tet}$ exhibited higher monomer conversion ratio (37% within 1 hour) compared to that of the engineered strains P$_{tac}$ and P$_{bad}$ (Fig. 4c). Hence, the elevated expression levels of the flavin biosynthetic cluster significantly increased flavins production, which consequently accelerated the *S. oneidensis*-triggered RAFT polymerization.

In addition, increased cell membrane permeability could facilitate flavins transport across cell membrane and secretion outside of *S. oneidensis* cells, thus enabling higher flavins level in the polymerization solution. The pore protein OprF from *Pseudomonas aeruginosa* was thus incorporated into the recombinant *S. oneidensis* strain to enhance flavins transport. To prevent expression of the porin protein that caused cytotoxicity, we designed several ribosome binding sites (RBSs) to regulate the expression level of OprF (Fig. 4a, and Supplementary Table 9). RBS is an effective regulatory element to initiate

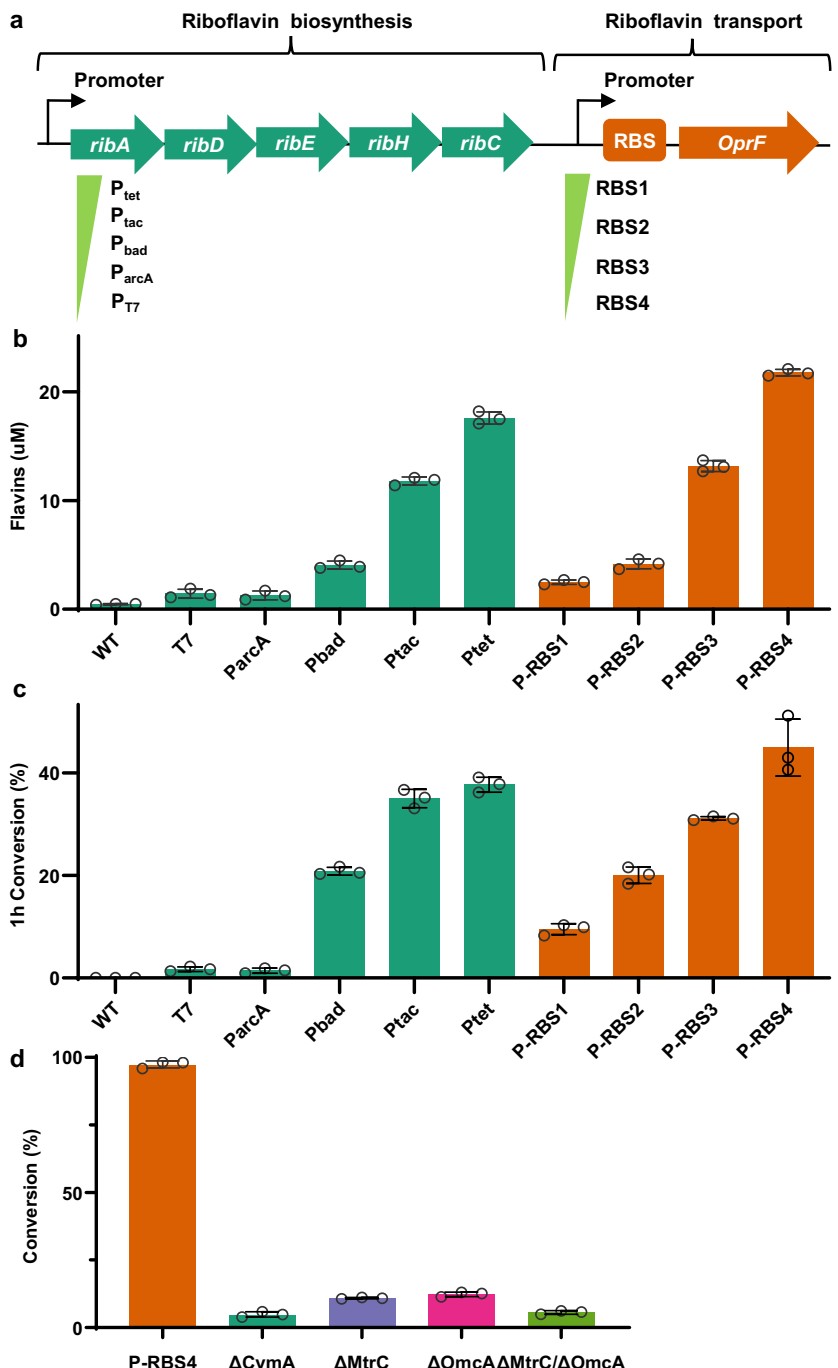

**Fig. 4 | Genetic engineering of *S. oneidensis* for the design and optimization of the de novo biosynthesis and transport of flavins, and its impact on the monomer conversion ratio of the RAFT polymerization. a** Plasmid map of the plasmid expressing the flavin biosynthesis gene cluster (*ribADEHC*) under the control of different inducible promotors ($P_{tet}$, $P_{tac}$, $P_{bad}$, $P_{arcA}$), and the porin gene *OprF* under the control of different ribosomal binding sites (RBS). RBS1: BBa_B0034, iGEM; RBS2: BBa_B0030, iGEM; RBS3: BBa_B0032, iGEM; RBS4: BBa_B0031, iGEM. **b** Quantification of flavins produced by the genetically engineered *S. oneidensis* harboring different promotors and RBS, respectively. **c** Comparison of monomer conversion ratio in the RAFT polymerization by the engineered *S. oneidensis* strains with different promoters at optimal inducer concentrations and optimal RBS, respectively. **d** The effect of gene knockouts of *c*-Cyts on the monomer conversion ratio of the *S. oneidensis*-triggered RAFT polymerization. Data showed mean ± SD of three independent experiments. Data in (**b**–**d**) were shown as the mean ± SD ($n = 3$). Source data were provided as a Source Data file.

DNA translation, whose sequence affects the translation initiation rate[77]. We observed that the decrease in the RBS strength increased the extracellular level of flavins (as shown in Fig. 4b, and Supplementary Fig. 22). The strong RBS1 that controlled the strong translation of the OprF porin protein decreased the flavins biosynthesis level (2 µM) and weakened the conversion ratio (9%) of the polymerization compared to that of the strain $P_{tet}$ (conversion ratio of 37%). This was probably because overexpression of porin protein OprF was toxic to the cells and reduced cell growth. We then identified a weak RBS4 in the recombinant *S. oneidensis* strain P-RBS4 that led to a higher flavins concentration (21.7 µM) and polymerization conversion rate (44%) than that of the strain $P_{tet}$ (Fig. 4c). These results suggested that

optimizing expression of the porin protein OprF accelerated flavins transport across the cell membrane to increase extracellular flavins concentration, consequently increasing the monomer conversion ratio of the polymerization. Through RT-qPCR analysis (Supplementary Fig. 23), we found that the transcription levels of the genes *ribA*, *ribD*, *ribE*, *ribH*, *ribC*, and *OprF* were upregulated in strain P-RBS4 compared to that of the WT. Furthermore, to assess the kinetic of the P-RBS4-mediated polymerization, a linear correlation of conversion ratio vs. reaction time and MWD vs. conversion was observed, suggesting the living character of the P-RBS4-triggered polymerization (Supplementary Fig. 24, and Supplementary Table 10).

The intracellular electrons generated from the metabolism of D-lactate flow through *c*-type cytochromes (*c*-Cyts) CymA, MtrA and MtrB to reach the outer membrane *c*-Cyts, OmcA and MtrC, forming the EET pathway of *S. oneidensis*. Subsequently, the electrons diffuse to and reduce flavins (the electron shuttles) to form the reduced form of $FL_{hq}$[58,76]. To understand the impact of the EET components (i.e., *c*-Cyts) on RAFT polymerization, we further studied the effect of knocking out the selected *c*-Cyts on the polymerization kinetics. The ΔOmcA or ΔMtrC knockout strain lacked the efficient EET pathway led to significantly reduced polymerization activity (conversion ratio <12%) in comparison to the strain P-RBS4 (Fig. 4 d), supporting the participation of *S. oneidensis* in discharging electrons to flavins. Specifically, the double genes-mutant strain with knockout of MtrC and OmcA (ΔMtrCΔOmcA) showed dramatically attenuated monomer conversion ratio (5%). Additionally, the inner membrane *c*-Cyt (ΔCymA) knockout strain exhibited very low polymerization activity (monomer conversion ratio 4%). However, complete abolishment of polymerization activity was not observed because other *c*-Cyts might perform the complementary function of EET in *S. oneidensis*. Taken together, these results highlighted the critical role of *c*-Cyts in delivering electrons from bacterial cells to extracellular flavins, which can be effectively modulated through genetic modifications.

To better understand the reaction process of the *S. oneidensis*-triggered RAFT polymerization, its reaction mechanism is analyzed. The *S. oneidensis* cells played an essential role in initiating and sustaining the RAFT polymerization reaction system: (i) as a living electrode or electron donors to continuously supply electrons to sustain the polymerization process; (ii) as an efficient flavins (the electrophotocatalyst) producer to obviate the need of external flavins addition; and (iii) as an oxygen scavenger to eliminate dissolved oxygen in the reaction mixture to reduce radicals quenching[37]. Subsequently, the flavins acted as an electron relay, which shuttled the intracellular electrons from the cells to CTAs. The electron transfer pathways in the radical polymerization can be modularized into three steps (Fig. 1): (i) production of intracellular electrons via lactate metabolism by the *S. oneidensis* cells; (ii) flavins reduction to form $FL_{hq}$ via EET; and (iii) CTA reduction by $FL_{hq}^{*}$ to form radicals via PET. Based on the electron transfer pathways, we proposed a mechanism of the *S. oneidensis*-triggered RAFT polymerization (as shown in Fig. 1). Four electron equivalents per lactate molecule were generated intracellularly in *S. oneidensis*, which were transferred through the inner membrane *c*-Cyt (CymA), periplasmic *c*-Cyt (MtrA) and outer membrane *c*-Cyts (MtrC and OmcA) to the endogenously secreted electron mediators flavins. Bioreduction of the yellow-colored flavins via microbial EET generated the fully reduced colorless state of $FL_{hq}$. Finally, photoexcitation of $FL_{hq}$ led to the generation of a strong reducing intermediate $FL_{hq}^{*}$, which can reduce CTAs via PET to form free radicals to initiate radical polymerization with concomitant generation of flavins, thus completing the biocatalytic cycle.

## Monomer diversity and block copolymer synthesis in *S. oneidensis*-triggered RAFT polymerization

To broaden the scope, we subsequently explored the compatibility of the *S. oneidensis*-triggered RAFT polymerization with a variety of monomers and CTAs (Table 1). Firstly, utilizing 2-ethylsulfanylthiocarbonylsulfanyl-propionic acid methyl ester (CTA1) as the thiocarbonylthio CTA, the water-soluble monomers including poly (ethylene glycol) methyl ether acrylate (PEGA, average $M_{n}$ = 480 g mol⁻¹), 2-hydroxyethyl methacrylate (HEMA), methacrylatoethyl trimethyl ammonium chloride (TMAEMA), 2-aminoethyl methacrylate hydrochloride (AOMA), 4-acryloylmorpholine (AML), N-isopropylacrylamide (NIPAM), and sodium methacrylate (MAA), were examined. The RAFT polymerization for most of the monomers attained high polymerization conversion ratio (> 93% within 6 h polymerization) and favorable dispersities (Đ < 1.23), except for the monomer MAA (Supplementary Figs. 31–39). For the monomer MAA, the conversion ratio can only reach a moderate level (86%), possibly due to the high concentration of MAA causing a decrease in the pH of the polymerization system. Notably, for the synthesis of the specific functional polymer, we successfully synthesized the temperature-sensitive polymer PNIPAM (23.2 kg mol⁻¹, Đ = 1.15) with a lower critical solution temperature (LCST), which was measured to be 27 °C. Additionally, the water-insoluble monomer methyl methacrylate (MMA) could also be polymerized via the emulsion polymerization with a near-quantitative conversion ratio (99%) and low dispersity (Đ = 1.25).

We further examined application of the *S. oneidensis*-triggered RAFT polymerization to synthesize PDMA polymers using a few CTAs with different reduction potentials. Relatively good control over molecular weight and low dispersity were obtained for 4-cyano-4-[(ethylsulfanylthiocarbonyl)sulfanyl] pentanoic acid (CTA2) and *S*,*S*′-bis(α,α′- dimethyl-α″-acetic acid) trithiocarbonate (CTA3), except for ethyl 2-ethoxycarbothioylsulfanylacetate (CTA4) with a high dispersity (Đ = 2.76) which was attributed to its lower chain-transfer kinetic constant[78]. Taken together, these results suggested that the polymerization is suitable for a variety of monomers and CTAs.

In addition, based on the terminal fidelity and the range of monomers compatible with the *S. oneidensis*-triggered RAFT polymerization, we further investigated the synthesis of block copolymers (Supplementary Table 7). A PDMA macroinitiator was initially synthesized using CTA1. Then, adding monomer AML in situ, the target diblock copolymer PDMA-*b*-PAML ($M_{n,GPC}$ = 14.7 kg mol⁻¹, Đ = 1.24) can be obtained (Supplementary Fig. 25). Similarly, the chain extension of PDMA was also successful with PEGA to yield well-defined PDMA-*b*-PPEGA ($M_{n,GPC}$ = 18.5 kg mol⁻¹, Đ = 1.20) (Supplementary Fig. 26). In both cases, the GPC traces showed a clear shift of the MWDs with minimal tailing in the homopolymer region. Furthermore, using CTA2 as the chain-transfer agent, the chain extension and block copolymers were also investigated with this polymerization system. As predicted, the chain extensions of the macroinitiator with DMA were successful, yielding deblock polymer PDMA-*b*-PDMA and triblock polymer PDMA-*b*-PDMA-b-PDMA with increased molecular weight from 21.8 kg mol⁻¹ to 30.6 kg mol⁻¹ (Supplementary Fig. 27). And block copolymerization products (PDMA-*b*-PAML and PDMA-*b*-PPEGA) can also be synthesized, which all displayed controlled dispersities (Đ = 1.13–1.16) as supported by the shifts of the GPC curves (Supplementary Figs. 28 and 29). Taken altogether, these results manifested the versatility of the *S. oneidensis*-triggered RAFT polymerization.

However, due to the complex physiological activity of microbial cells and their responses to external stimuli, the rational design of functional polymer materials via microorganism-triggered RAFT polymerization remains a significant challenge. High-throughput polymer synthesis and screening emerged as an attractive alternative strategy, enabling the rapid identification of structure-property relationships and the discovery of novel materials[46]. However, one of the key challenges in high-throughput polymerization was to conduct efficient reactions in small volumes and under ambient conditions. Our *S. oneidensis*-triggered RAFT polymerization system had the feature of oxygen tolerance and high polymerization efficiency, which were thus explored to facilitate high-throughput polymer synthesis.

**Table 1 | The *S. oneidensis*-triggered RAFT polymerization of various monomers and CTAs**

Polymerization conditions: After 12 hours' culturing of the engineered strain PRBS4, the concentration of cells in the culture medium was measured to be [OD$_{600}$] = 1.5, and the flavins concentration was 21 μM. Then, take a certain amount of the bacterial culture, monomer, CTA, and M9 medium to form a 2 mL polymerization reaction mixture. The final concentrations were [cell OD$_{600}$] = 1, [Flavins] =14 μM, [Monomer] = 10% (w/v) and [CTA] = X mM (calculated based on the ratio of monomer to CTA). The polymerization reaction mixture was purged with nitrogen for 5 minutes and exposed to blue LED light (10 W, 460 nm) at room temperature. Conversion ratio was determined by $^1$H NMR spectroscopy. Molecular weight ($M_n$) and polydispersity (Đ) were determined by GPC analysis.

To assess whether RAFT polymerization could proceed efficiently without deoxygenation in small-volume systems, we initially evaluated the polymerization kinetics of DMA in 96- and 384-well microtiter plates, with reaction volumes of 200 μL and 50 μL, respectively. Under blue light illumination, all polymerizations reached high conversion ratios with low dispersity (Fig. 5a). The polymerization kinetics exhibited a linear correlation of ln([M$_0$]/[M]) versus reaction time, indicating the livingness of the well plate polymerization (Supplementary Fig. 30). Notably, there was no significant difference in polymerization rates between the 50 μL and 200 μL polymerization reactions (Supplementary Fig. 30), demonstrating the robustness of the system across various scales. To further demonstrate the applicability of *S. oneidensis*-triggered RAFT polymerization for synthesizing polymers with a range of molecular weights, we constructed a PDMA library targeting degrees of polymerization (DPs) from 100 to 2000 in a 96-well format. As shown in Fig. 5, all polymerizations achieved high monomer conversions. The molecular weight dispersity showed a slight increase with the growth of polymer molecular weight, which was maintained at a well-controlled fashion even at high DPs (Đ < 1.27) (Supplementary Table 11). GPC characterization showed a monomodal peak with a small tail, confirming that well-defined polymers with a wide range of molecular weights can be achieved. Taken together, these results suggested that the *S. oneidensis*-triggered polymerization system provided a robust and efficient platform for high-throughput RAFT polymerization, offering a promising approach for the rapid synthesis and screening of polymer libraries.

In conclusion, we developed a *S. oneidensis*-triggered RAFT polymerization system that was able to avoid end-group heterogeneity via directly reduction of CTAs by microorganisms and light irradiation to continuously generate radicals to initiate and sustain RAFT polymerization. This microorganism-triggered RAFT polymerization system used genetically engineered *S. oneidensis* to synthesize and secrete flavins (riboflavin and FMN) as the electron mediators, harnessed lactate metabolism of *S. oneidensis* to generate electrons and the extracellular electron transfer pathway of *S. oneidensis* to regenerate the fully reduced flavins hydroquinone (FL$_{hq}$), and utilized the concomitant visible light irradiation to excite the reduced form FL$_{hq}$ to form the photoexcited FL$_{hq}^*$, which consequently transferred an electron to CTAs via photoinduced electron transfer, ultimately generating radicals to initiate RAFT polymerization. Mechanistically, the RAFT polymerization reaction proceeded through a photo-excited extracellular electron transfer process that combined bioreduction and photoexcitation, utilizing flavins as the electron mediator to promote electron transfer from the cell to the CTA. This dual catalytic mechanism overcame the thermodynamic hurdle associated with the CTA reduction by cell-secreted flavins, ensuring a steady supply of radicals and suppressing end-group heterogeneity and chain termination of the RAFT polymerization reaction. The engineered *S. oneidensis*-triggered RAFT polymerization was successfully applied to various monomers and CTAs to produce a range of homopolymers, block copolymers with high monomer conversion ratio (as high as 99%) and low dispersity (Đ as low as 1.11). We believe the

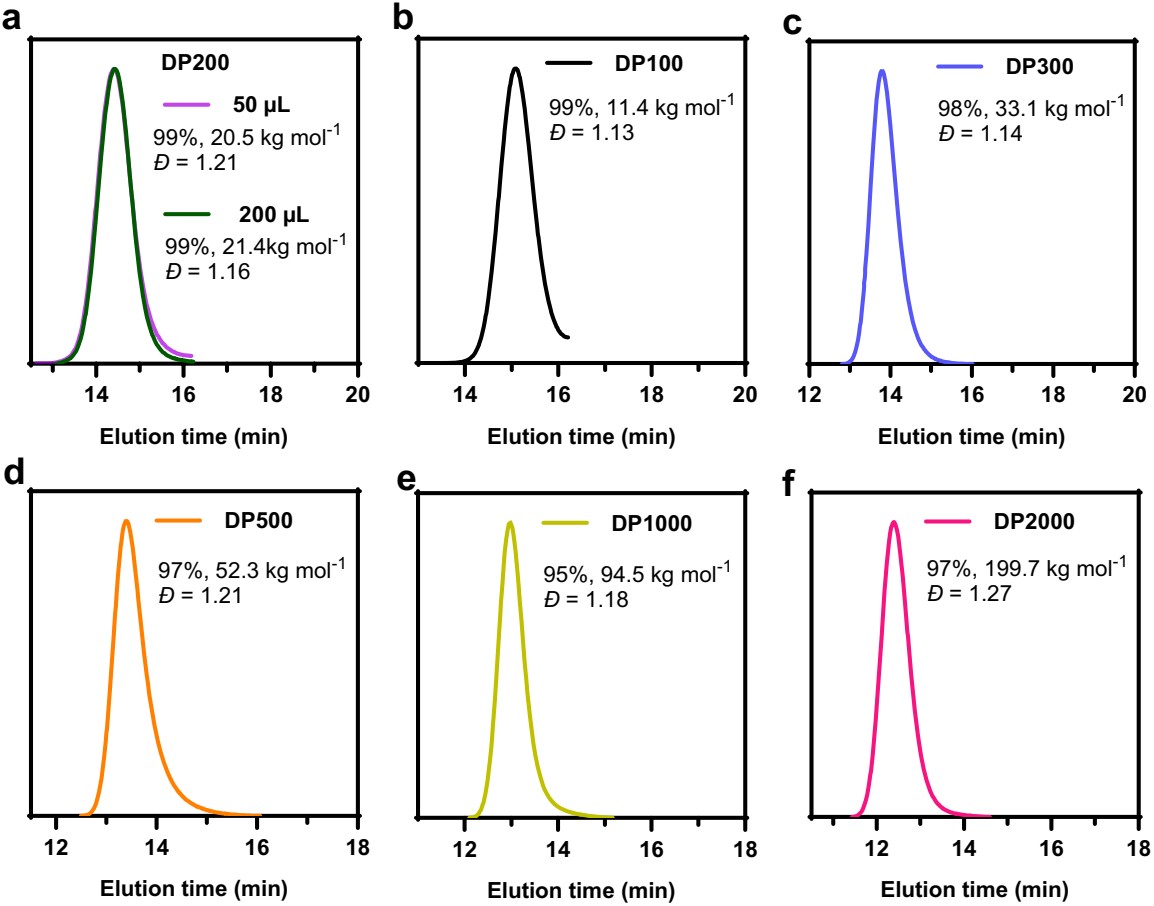

**Fig. 5 | GPC (Gel permeation chromatography) traces of high-throughput *S. oneidensis*-triggered RAFT polymerization. a** Polymers PDMA (DP200) obtained from the corresponding 96 and 384-well plates. **b–f** The PDMA library targeting degrees of polymerization (DPs) from 100 to 2000 in a 96-well plate. **b** DP100. **c** DP300. **d** DP500. **e** DP1000. **f** DP2000. Source data were provided as a Source Data file.

microorganism-triggered RAFT polymerization system could be as a powerful means to achieve living microorganisms-mediated RAFT polymerization, as well as expanding to many living organisms.

## Methods

### General conditions of electroactive microorganism-triggered polymerization

Homogeneous polymerizations were conducted via the following general procedure. For most experiments, [riboflavin] or [FMN] = 5 μM, *S. oneidensis* [$OD_{600}$] = 1, [monomer] = 1 M, [CTA] = 2 mM, [DMSO] = 5% (v/v), unless otherwise stated. Each of the reagents for polymerization were then added into the M9 culture to a 10 mL reaction flask to make a final solution 2 mL. The vessel was sealed, covered with aluminum foil, and purged with nitrogen for 5 minutes. The reaction solution was then illuminated blue LED light (10 W, 460 nm) at room temperature. Samples were taken periodically to monitor monomer conversion ratio and molecular weight dispersity of the resulting polymer using ¹H NMR and gel permeation chromatography (GPC) analysis, respectively.

### Synthesis of block copolymers

Block copolymers were prepared using a one-pot procedure, where the first block was synthesized to complete monomer conversion before adding the second monomer. The synthesis of block polymer was described below as a representative procedure. In the synthesis of the first block polymer, each of the reagents were added to a 10 mL reaction flask. After purging the solution with nitrogen for 5 min,

polymerization was conducted at room temperature under blue LED light. After 6 h of polymerization, the vessel was transferred into the anaerobic glove box and the second monomer was added to the first block polymer solution. After 6 h of blue light exposure, an aliquot was withdrawn for analyses by GPC and ¹H NMR spectroscopy.

### Polymerization procedure for light on-off experiment

Light on-off experiments of microorganism-triggered RAFT polymerizations were conducted via the following general procedure. [riboflavin] = 5 μM, *S. oneidensis* $OD_{600}$ = 1, [DMA] = 0.5 M, [CTA] = 2 mM and a certain amount of M9 culture were added to a 10 mL reaction flask to make a final solution 2.0 mL. The vessel was sealed, covered with aluminum foil, and purged with nitrogen for 5 min. The polymerization solution was then illuminated by blue LED light (10 W, 460 nm) at room temperature. The samples were irradicated for a set period of time, after which the vessel was transferred into the anaerobic glove box and an aliquot was taken for ¹H NMR analysis. The lights were switched off, and the reaction vial was analyzed again by ¹H NMR. The lights were switched back on, and the sample was irradiated for another period of time, followed by aliquot collection for further characterization. The light on-off cycle was repeated multiple times until monomer conversion ratio exceeded 95%.

### Characterization of synthesized polymers

Polymer samples were analyzed directly using gel permeation chromatography (GPC) with an inline differential refractive index (DRI)

detector, and nuclear magnetic resonance ($^1$H NMR) spectroscopy. For the water-soluble polymers, the GPC analysis was performed in a Shimadzu LC-20A GPC system using Milli-Q water with 0.5 mM NaNO$_3$ as eluent. The GPC system comprises a TSKgel guard column PW$_{XL}$ (6.0 mm I.D. × 4 cm) followed by a TSKgel GMPW$_{XL}$ (7.8 mm I.D. × 30 cm, 13 μm) and a refractive-index detector. The flow rate of water was 0.6 mL min$^{-1}$ and the test temperature was 35 °C. Molecular weight and dispersity values were determined with the Shimadzu software package and comparison with low dispersity poly(ethylene glycol) standards. For the water-insoluble polymers, the GPC analysis was conducted on a Waters 1515 GPC system using THF as eluent and polystyrene with narrow dispersity as standards. The GPC system comprises three Agilent columns (7.8 × 300 mm) and a refractive-index detector. The water flow rate was set to 0.6 mL/min and the test temperature was 35 °C. Nuclear magnetic resonance (NMR) spectroscopy was performed on a Bruker AVANCE III 400 MHz spectrometer for $^1$H and $^{13}$C, using D$_2$O, DMSO-d6 or CDCl$_3$ as the solvent.

## Conversion ratio calculation

Estimated conversion ratio was calculated by $^1$H NMR using the ratio of integrals for monomer: polymer peaks. For PDMA, the monomer CH$_2$ from the CH$_2$ acrylate peak (5.68-5.71 ppm) and CH$_2$ polymer backbone peak (1.29-1.54 ppm) were compared with Eq. (1)[50].

$$\alpha = \frac{\int_{protons}^{polymer}}{\int_{protons}^{polymer} + \int_{protons}^{monomer}} *100 \qquad (1)$$

The theoretical molecular weight ($M_{n,th}$) was calculated by Eq. (2)[50].

$$M_{n,th} = x = \frac{[DMA]_0}{[CTA]_0}*MW^{DMA}*\alpha + MW^{CTA} \qquad (2)$$

where $[DMA]_0$, $[CTA]_0$, $MW^{DMA}$, $\alpha$ and $MW^{CTA}$ correspond to DMA and CTA concentration, molar mass of DMA, monomer conversion ratio, and molar mass of CTA.

## Testing viability of bacteria

To quantify viable bacterial concentration, a count of colony forming units (CFU) was undertaken. 100 μl of bacterial culture was diluted in PBK and then plated on LB agar plates to determine the CFU mL$^{-1}$. The plates were incubated at 30 °C overnight and colonies were manually counted.

## Culture conditions and flavins measurement

Overnight cultures of *S. oneidensis* MR-1 (harboring the empty vector pYYD, Pbad, Ptac, Ptet, and ParcA, respectively) was transferred to M9 broth at a ratio of 1:100, supplemented with 50 μg mL$^{-1}$ kanamycin and various concentrations of inducer. After approximately 12 h of growth at 30 °C with shaking at 200 rpm, flavin levels were quantified.

## Determination of flavins

Flavins were measured by using a Shimadzu LC-20A GPC system with a UV detector. All standard solutions and samples supernatants were filtered and analyzed using a reverse-phase C18 column (5 μm particle size, 250 × 4.6 mm$^2$, Thermo Fisher Scientific). Chromatographic separations were done using a mobile phase of methanol and 0.01 M NaH$_2$PO$_4$ at 30 °C with an elution rate of 0.6 ml min$^{-1}$. Flavins concentrations were determined by monitoring signals at 445 nm.

## In vitro gene synthesis, plasmid construction and transformation

The *ribADEHC* genes from *B. subtilis* and the *OprF* gene from *P. aeruginosa* were identified, and their coding sequences were retrieved from KEGG database. Codon optimization for expression in *S.*

*onedensis* was performed using the online tool (http://www.jcat.de/), avoiding EcoRI, XbaI, SpeI, SdaI (Sbfl) restriction sites. The finalized sequences were synthesized in vitro (GENEWIZ, China), featuring EcoRI and XbaI at the upstream prefix, an RBS site, and SpeI and SdaI at the downstream suffix. The promoter P$_{bad}$, P$_{tac}$, P$_{tet}$, P$_{arcA}$, along with the *ribADEHC* genes, were also synthesized in vitro (GENEWIZ, China). These constructs were individually cloned into the PYYD vector, which has been created earlier by our group. The resulting plasmids were introduced into the dap-auxotrophic strain *E. coli* WM3064 and subsequently transferred into *S. onedensis* via conjugation. To support *E. coli* WM3064 growth, 100 μg mL$^{-1}$ 2, 6-diaminopimelic acid (DPA) was provided.

## Construction of *S. oneidensis* mutant strains

The procedure for gene deletion is briefly described. The gene knockout of *c*-type cytochromes in *S. oneidensis* MR-1 was performed using a suicide vector-based strategy, as previously described[79]. The left and right homology arms flanking the target gene-each containing the *attB* site-were PCR-amplified from the genome and subsequently fused via overlap extension PCR. The fused fragment was cloned into the suicide plasmid pHG1.0 using the Gateway BP Clonase enzyme mix (11789021, Invitrogen, USA). The resulting construct was transferred into *S. oneidensis* MR-1 by conjugation. Transconjugants carrying the first homologous recombination event were selected using gentamicin (15 μg/mL) and verified by colony PCR. These single-crossover integrants were then plated on LB agar supplemented with 10% sucrose to select for the second recombination event, which resulted in the excision of the plasmid and successful deletion of the target gene. Final deletion mutants were confirmed by PCR and Sanger sequencing. This method was used to generate single and multiple knockouts of key *c*-type cytochromes, including *mtrC*, *omcA*, and *cymA*.

## Polymerization conditions of RAFT triggered by engineered *Shewanella oneidensis* strain

Homogeneous polymerizations were conducted via the following general procedure. After 12 h of culturing the engineered strain, the concentration of cell in the culture fluid was measured. Then, a certain amount of the culture fluid containing the bacteria was taken to achieve a final concentration of $[OD_{600}] = 1$ in the polymerization reaction mixture. The reagent of [DMA] = 1 M, [CTA] = 2 mM, the culture fluid and a certain amount of M9 culture were added to a 10 mL reaction flask to make a final solution 2 mL. The vessel was sealed, covered with aluminum foil, and purged with nitrogen for 5 min. The reaction solution was then illuminated blue LED light (10 W, 460 nm) at room temperature.

## Electrochemical measurements

Cyclic voltammetry experiments were conducted using a CHI660D electrochemical workstation (Shanghai) with a one-compartment electrolysis cell. The setup included a typical glassy carbon working electrode (3 mm diameter), a platinum wire counter electrode, and an Ag/AgCl (KCl saturated) reference electrode, all at 25 °C. The measurements were performed in acetonitrile with 0.1 M tetra-butylammonium as the supporting electrolyte, at a scan rate of 50 mV s$^{-1}$, or in PBK (0.1 M) aqueous solution under the same scan rate.

## UV-vis absorption spectroscopy and fluorescence spectroscopy

UV-vis absorption spectra and fluorescence spectra were measured on a Thermo Scientific Varioskan LUX multimode microplate reader.

## Fluorescence lifetime measurement by time-resolved fluorescence spectroscopy

All samples were prepared in a MBraun glovebox with oxygen levels <0.1 ppm. Each sample was prepared in 100 mM PBK buffer pH7.5 with 0.1 mM flavins. To reduce the flavins to hydroquinone form FL$_{hq}$, we

added 4 mM $Na_2S_2O_4$, followed by addition with and without 10 mM CTA. During the course of the experiment samples were stirred and translated continuously in a 2 mm quartz cuvette, which was capped to ensure no introduction of oxygen. The transient photoluminescence measurement was conducted on a time-resolved spectrometer equipped with a time correlated single photon counting (TCSPC) module (HiLight 990, Oriental Spectra). A picosecond 405 nm laser (pulse width <250 ps, PLD405, Oriental Spectra) was used for optical excitation. The time traces were fitted to single or multi-exponential decays through re-convolution with the measured instrument response function (IRF).

### Electron spin resonance (ESR) spectroscopy

An ESR spectrometer (JEOL, JES-FA200) was applied to identify radicals generated during photo-reaction. The following parameters were used for ESR detection: Center field, 329 mT, sweep width 7.5 mT, microwave power, 1 mW, sweep time, 60 s, time constant, 0.03 ms, modulation frequency, 100 kHz, modulation amplitude, 2 mT. DMPO (20 μM), RF (5 μM), sodium dithionate (50 μM), CTA (1 mM) were added into PBK buffer. The vessel was sealed, wrapped with aluminum foil, and purged with nitrogen for 20 min, then irradiated under blue light at room temperature with stirring for 30 min. The solution was then quickly transferred to a capillary tube for ESR measurement.

### Computation details

The computations were carried out employing Gaussian 16 Rev. C.01, utilizing the Density Functional Theory (DFT) methodology for the ground state and Time-Dependent DFT (TD-DFT) for the excited state. Initially, an approximate optimization of the ground-state geometry was executed at the B3LYP/6-31 G level, incorporating a supplementary quadratic convergence threshold. Subsequently, the ground-state geometry underwent further refinement at the B3LYP/6-311 G + +(d,p) level, employing a force-minimization procedure and employing a self-consistent field convergence threshold of $10^{-10}$ atomic units. This optimization process encompassed the inclusion of bulk solvent effects through the polarizable continuum model, with the dielectric permittivity set to that of water. Subsequent to the geometry optimization, frequency calculations were conducted to verify that the obtained geometry corresponded to an energy minimum. This procedure was also used to calculate the thermodynamic correction of Gibbs free energy. Consequently, the first 25 lowest-energy excited states were determined by means of TD-DFT calculations at the CAM-B3LYP/6311 + G(2 d, p) level of theory. This calculation employed a stringent self-consistent field convergence threshold of $10^{-10}$ atomic units. All visual representations of the computed species were generated using the Gauss View software, facilitating the visualization of quantum chemistry computations. The change in free energy (ΔG) for each step in a reaction pathway is determined by the difference in the sum of energies between the reactants (or initial state) and products (or final state) of that step.

### Reporting summary

Further information on research design is available in the Nature Portfolio Reporting Summary linked to this article.

## Data availability

The data supporting the findings of this study are available within the paper and its Supplementary Information/Source Data files. A reporting summary for this article can also be found as a Supplementary Information file. Source data are provided with this paper.

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

## Acknowledgements

The authors acknowledge the financial support from the National Key Research and Development Program of China (2025YFA0921900 to H.S.), the National Natural Science Foundation of China (NSFC 22378305 and 32071411 to H.S.), and the Haihe Laboratory of Sustainable Chemical Transformations.

## Author contributions

C.L. and J.L. contributed equally to the work. C.L. and J.L. designed the project, performed experiments, analyzed data, and drafted the manuscript. W.H., L.X., F.L., Q.L., J.Z., H.Y., B.Z. helped to perform some experiments. D.X., S.C., W.L., and K.H.N. helped to revise the manuscript. H.S. designed and supervised the project, obtained research grants, analyzed data, and critically revised the manuscript.

## Competing interests

The authors declare no competing interests.
