## [Transparent Peer Review file · Nature Communications]

Photo-excited extracellular electron transfer of electroactive microorganism triggers RAFT polymerization

Corresponding Author: Professor Hao Song

Version 0:

Reviewer comments:

Reviewer #1

(Remarks to the Author)

The authors developed a bio-electro-photocatalytic system that leverages flavins from *Shewanella oneidensis* as the catalyst to directly reduce CTA, generating radicals for RAFT polymerization. This BioEPC system enables controlled radical polymerization without requiring external initiators or catalysts, which will benefit the living microorganisms-activated complex chemical reaction. There are some comments to be addressed before further consideration.

1. The writing needs improvements. There are some vague or long descriptions should be clarified, for example, (1) "To eliminate the need of exogenous initiators that may lead to cytotoxicity on cells and end-group heterogeneity of polymer products, and use the electrons continuously generated by cell metabolism of lactate (carbon source and electron donor) by *S. oneidensis* to provide radicals to enable microorganisms-mediated RAFT polymerization, we constructed a BioEPC-RAFT polymerization system, in which the electron shuttles flavins (riboflavin and FMN) synthesized by *S. oneidensis* played as the electro-photocatalyst to reduce the CTA upon light irradiation to produce radicals, thus initiating RAFT polymerization". (2) The article uses flavins as catalysts, sometimes referring to them as photo-electrocatalysts and other times as electro-photocatalysts. Please unify the terminology. (3) It should be flavins transports in Fig. 1. Therefore, carefully check the whole MS for better understanding.
2. In Fig. 2c, the authors calculated Gibbs energy profile of the electron transfer pathways from flavins to CTA1. However, they did not provide the calculation method, and optimal Gibbs energy value, and the energy used for reaction?
3. How to calculate the extracellular electron transfer and photoinduced electron transfer efficiency? If there is any method to calculate the ratios of electrons from microbe being used for CTA reduction?
4. As we know, some nanoparticles can help the EET or photoelectron uptake, can it be integrated to this system?
5. Although the authors provide the riboflavin concentration in Fig. 4b, it is required to present evidence if the riboflavin biosynthesis genes *ribADEHC* and *oprF* were expressed by qPCR or SDS-PAGE or Western blot?
6. Please provide the c-Cyts gene knockout/deletion design details.
7. The authors may provide more experiment data on the stability of BioEPC-RAFT polymerization system, for example, system size, pH, and temperature.
8. The authors proposed this controlled radical polymerization can be controlled by light ON/OFF, but how to maintain the activity or viability of *S. oneidensis* if they perform a long-term reaction?
9. In the Methods, how did the authors sample since they need to keep the oxygen limited condition?

Reviewer #2

(Remarks to the Author)

Song and co-workers report the development of a bio-electro-photocatalytic process to directly reduce the C-S bond of thiocarbonylthio RAFT agents to initiate controlled radical polymerisations. Their approach combines electrocatalytic bacterium and photoactive components such as riboflavin and flavin mononucleotide (FMN). They demonstrate that their RAFT process retains control over molar mass and MW distribution, and that this process can be temporally controlled depending on irradiation and concentrations of the reagents.

While this approach is certainly novel I do not believe this is a significant step forward for either living cell mediated RAFT polymerisation, which has been demonstrated via a few different approaches as described in the introduction. Neither is this a huge leap forward for photocatalytic RAFT polymerisation which has been extensively studied. The merger of these is

interesting, but does not warrant publication in such a prestigious journal. I think that this could be resubmitted elsewhere as the findings are sound, figures are presented well and the paper is written very well.

For this to be considered in a higher impact journal I would want to see an application which exploits the merger of the bio and photocatalytic RAFT process.

Finally, I also would recommend the authors to carefully explain how this new RAFT process can be distinguished from traditional iniferter RAFT polymerisation, as that is also a photomediated process but does not need bacteria or associated photo-catalytic molecules hence one could argue that the presented bio-photocatalytic approach is over complex.

Therefore I do not recommend this to be published in nature communications and the authors consider more field-specific journals.

Reviewer #3

(Remarks to the Author)

This manuscript reports an *S. oneidensis*-activated photo-RAFT polymerization process. It utilizes genetically engineered *S. oneidensis* to synthesize and secrete flavins (riboflavin and FMN) as photo-initiators. The riboflavin-photoactivated RAFT process is known. The novel aspect of this work is that the photocatalyst is produced via a bioprocess.

There are two major questions here:

1. Do you need the cells after the FLbq is formed? Can you kill the cells and still conduct the polymerization?
2. Is this process oxygen-tolerant? Why is it, or why is it not?

Therefore, I am not entirely sure if this is truly a microorganism-activated RAFT process or merely a microorganism-produced photo-initiator system. I think this distinction is a major issue that must be clarified before accepting the claim in this paper. Can you regulate cell growth to control the polymerization process? What are the experimental data supporting this?

Thus, it is premature to accept the paper with its current claim at this stage.

Version 1:

Reviewer comments:

Reviewer #1

(Remarks to the Author)

The authors performed additional experiments and provided supporting data to address the reviewers' comments. The current manuscript exhibited highly valuable information and will benefit the readers. Therefore, I think this manuscript can be accepted.

Reviewer #2

(Remarks to the Author)

Song and co-workers have resubmitted this manuscript following an initial round of review. The manuscript has been modestly revised with a two key elements. Firstly they have clarified the main novelty of the manuscript which was unclear previously.

They state that this novelty arises from being the first living cell initiated RAFT polymerisation where the radicals and photomediator are generated directly from the bacterial cells - which I agree with that this is the first time I have seen a system like this.

Secondly they have conducted further experiments demonstrating low volume and open-to-air polymerisations in 96 well plates, potentially showing how this system could be utilised in other applications.

Despite these changes and assertion of novelty, I feel that the advancements here are relatively niche and the extra experiments do not fully justify this relatively complex system.

For instance it has already been shown that bacterial mediated polymerisations can be done under non-inert conditions using exogenous initiators, either azo-aryl based like from Qiao and co-workers, or using a Fenton polymerisation from Gurnani and Rawson - ACS Macro Lett. 2022, 11, 8, 954–960. Hence the main novelty that remains is the ability to generate the flavin molecule from the engineered bacteria themselves. If they had shown that this is important in an application - e.g. for materials synthesis or processing, then I would understand.

I therefore recommend that this could be transferred to communications chemistry or more specialised journal as the impact of this work is not strong enough for broad readership.

A few minor comments -

Fig 1. is a bit confusing, the end product seems to be in the top left which is where i would anticipate that the start of the 'flow' would be - so i suggest redrawing this

Fig 3d - the trithiocarbonate groups are missing one sulfur atom in two of the polymer structures

I would recommend that the data for SI table 17 is put in the manuscript to show the SEC curves and NMR spectra.

Reviewer #3

(Remarks to the Author)

The authors have taken my comments seriously and conducted a number of experiments to address my questions. I am convinced that the polymerization process can be enhanced by modulating cell growth, and that this is part of the mechanistic cycle. The paper is now in an acceptable form.

Point-by-point response to the Reviewers' Comments:

Reviewer #1 (Remarks to the Author):

The authors developed a bio-electro-photocatalytic system that leverages flavins from *Shewanella oneidensis* as the catalyst to directly reduce CTA, generating radicals for RAFT polymerization. This BioEPC system enables controlled radical polymerization without requiring external initiators or catalysts, which will benefit the living microorganisms-activated complex chemical reaction. There are some comments to be addressed before further consideration.

Response: We sincerely thank the Reviewer for the valuable suggestions! The manuscript has been carefully revised according to these suggestions and comments.

1. The writing needs improvements. There are some vague or long descriptions should be clarified, for example, (1) "To eliminate the need of exogenous initiators that may lead to cytotoxicity on cells and end-group heterogeneity of polymer products, and use the electrons continuously generated by cell metabolism of lactate (carbon source and electron donor) by *S. oneidensis* to provide radicals to enable microorganisms-mediated RAFT polymerization, we constructed a BioEPC-RAFT polymerization system, in which the electron shuttles flavins (riboflavin and FMN) synthesized by *S. oneidensis* played as the electro-photocatalyst to reduce the CTA upon light irradiation to produce radicals, thus initiating RAFT polymerization". (2) The article uses flavins as catalysts, sometimes referring to them as photo-electrocatalysts and other times as electro-photocatalysts. Please unify the terminology. (3) It should be flavins transports in Fig. 1. Therefore, carefully check the whole MS for better understanding.

Response: We sincerely appreciate the Reviewer for the thoughtful and detailed comments regarding the clarity, terminology, and consistency of our manuscript. We have addressed each of the issues raised by the Reviewer, as shown below:

(1) Clarification of the long sentence: We agree that the sentence highlighted by the Reviewer was overly long and potentially confusing. We have revised it to improve clarity, readability, and logical flow. The revised version now reads:

"In traditional RAFT polymerization, exogenous radical initiators may cause cytotoxicity to cells and lead to end-group heterogeneity in polymer products. Thus, to eliminate the need for exogenous addition of radical initiators, we constructed a new electroactive microorganism-triggered RAFT polymerization system, in which *S. oneidensis*, an extensively studied model exoelectrogen, metabolizes lactate (serving as both the carbon source and the electron donor) to continuously generate electrons to trigger radical polymerization. In this system, flavins (riboflavin and FMN) synthesized by *S. oneidensis* act as the electron mediators."

(2) Terminology unification: We appreciate the Reviewer for pointing out the inconsistency in the terminology in describing the catalytic function of flavins. After careful consideration of the suggestions from the three Reviewers, we have standardized the terminology throughout the manuscript to “electron mediators”, which more accurately describes the role of flavins in our system. Specifically, flavins function as electron shuttles that transfer electrons derived from microbial metabolism to CTA under photoexcitation, thereby enabling radical generation in microorganism-triggered RAFT polymerization.

(3) Correction in Figure 1 label: We appreciate the Reviewer’s suggestion. The label in Figure 1 has been corrected from “flavins transport” to “flavins transport”. In addition, to more clearly reflect the underlying polymerization process, the term “BioEPC-RAFT” has been revised to “*S. oneidensis*-triggered RAFT,” and “bioelectro-photocatalysis” has been revised to “photo-excited extracellular electron transfer,” as shown below:

Figure 1. Construction of the electroactive microorganism-triggered reversible addition-fragmentation chain-transfer (RAFT) polymerization system by photoexcited extracellular electron transfer of the genetically engineered *Shewanella oneidensis*. First, the engineered *S.*

oneidensis was constructed by synthetic biology approach to heterologously express the flavins synthesis pathway from *Bacillus subtilis* and the pore protein OprF from *Pseudomonas aeruginosa* in *S. oneidensis*, which enhanced flavins biosynthesis and transportation. Second, flavins acted as the electron mediators that integrated microbial extracellular electron transfer (EET) and photoinduced electron transfer (PET) to reduce the chain transfer agent (CTA, i.e., thiocarbonylthio) for generation of free radicals, which triggered the microorganism-mediated RAFT polymerization. In this process, the electron transfer pathways included three steps: (i) production of electrons by engineered *S. oneidensis* through D-lactate metabolism via intracellular electron transfer process; (ii) flavins reduction via the EET process; and (iii) CTA reduction via PET. Specifically, the electrons produced by metabolizing D-lactate in *S. oneidensis* transferred via the cytoplasmic membrane protein CymA, which reduced periplasmic proteins that transferred electrons to the Mtr-pathway (i.e., electrons transferred through MtrA to MtrC and OmcA). Then, electrons transferred to the endogenously secreted flavins. Bioreduction of flavins (FL) via EET generated the fully reduced flavins hydroquinone (FL_{hq}). Photoexcitation of FL_{hq} led to the formation of a photoexcited intermediate FL_{hq}^{*}, which reduced CTA via PET to produce free radicals, initiating the RAFT polymerization.

(4) Full-manuscript proofreading: In accordance with the Reviewer's comment, we have performed a comprehensive review and revision of the entire manuscript and Supplementary Information. We carefully corrected vague or overly long descriptions, eliminated inconsistent terminology, and improved the overall readability and presentation of the text. We believe these revisions significantly enhance the clarity of our manuscript.

2. In Fig. 2c, the authors calculated Gibbs energy profile of the electron transfer pathways from flavins to CTA1. However, they did not provide the calculation method, and optimal Gibbs energy value, and the energy used for reaction?

Response: We appreciate the Reviewer for this valuable and insightful comment. We apologize for the omission of methodological details regarding the Gibbs free energy calculations presented in Fig. 2c. We have now included a detailed description of the calculation procedure and associated parameters in the revised Supplementary Information (see **Supplementary Information**, computation details, and **Supplementary Tables 8-9**).

3. How to calculate the extracellular electron transfer and photoinduced electron transfer efficiency? If there is any method to calculate the ratios of electrons from microbe being used for CTA reduction?

Response: We sincerely thank the Reviewer for the insightful question regarding the electron transfer efficiency in our system. Below is a detailed explanation on how the

photoinduced and microbial electron utilization efficiencies were calculated.

The extracellular electron transfer efficiency, referred to as coulombic efficiency (C_E), is the ratio of the coulombs actually recovered as current to maximum possible coulombs if all substrate is consumed to produce current. The C_E was calculated over a period of batch cycles using Eq. 1 ¹:

$$C_E = \frac{\text{Coulombs recovered}}{\text{Total coulombs in substrate}} = \frac{\int_0^{t_b} I dt}{F b_{ES} \Delta n} \quad (\text{Eq. 1})$$

where F is Faraday's constant (98,485 C mol⁻¹ of electrons), I (A) is the current, t_b (s) is the time period of a batch cycle, b_{ES} is the stoichiometric number of moles of electrons produced per mole of substrate ($b_{ES} = 4$ when lactate was used as the electron donor), and Δn (mol) is the change of substrate amount over a batch cycle.

The photoinduced electron transfer (PET) efficiency was calculated based on fluorescence quenching measurements using the following Eq. 2 ²:

$$\eta_{PET} = 1 - \frac{I}{I_0} \quad (\text{Eq. 2})$$

where η_{PET} is the photoinduced electron transfer efficiency; I_0 is the fluorescence intensity (or lifetime) of the donor fluorophore in the absence of the electron acceptor; I is the fluorescence intensity (or lifetime) in the presence of the electron acceptor.

To investigate the PET process involved in our system, we measured the fluorescence lifetimes of reduced flavins (FL_{hq}) with and without CTA using time-resolved fluorescence spectroscopy. Specifically, the oxidized state flavins were reduced using sodium dithionite (Na₂S₂O₄) as the electron donor and excited at 405 nm. In the absence of CTA1, the fluorescence lifetimes of RF_{hq} and FMN_{hq} were 3.52 ns and 3.38 ns, respectively (see Supplementary Figs. 9 and 10). Upon addition of CTA1, the lifetime decreased to 3.01 ns for RF_{hq}, and 2.23 ns for FMN_{hq}. Using Eq. 1, the calculated photoinduced electron transfer efficiencies were approximately 15% for RF_{hq}, and 34% for FMN_{hq}.

In our polymerization system, quantifying the exact proportion of microbial electrons used specifically for CTA reduction in our system is indeed challenging, as the electrons generated by microbial metabolism may be distributed among multiple pathways, including the maintenance of cellular functions, reduction of endogenous electron acceptors, losses during the photoinduced electron transfer process, as well as consumption during chain transfer and termination events in the polymerization process.

In principle, this ratio could be estimated by calculating the theoretical number of electrons required for the polymerization reaction and comparing it to the total number of electrons that could be generated from complete oxidation of the consumed carbon source (lactate in our system). This can be expressed as Eq. 3 ³:

$$\text{Electron Utilization Ratio} = \frac{n_{e,polymerization}}{n_{e,carbon\ source}} \quad (\text{Eq. 3})$$

where $n_{e,polymerization}$ is the total number of electrons theoretically needed to reduce CTA; $n_{e,carbon\ source}$ is the maximum number of electrons that can be produced from complete metabolism of the carbon source (e.g., lactate, which generates four electron equivalents per molecule of lactate,) consumed during the reaction.

The mechanism of RAFT polymerization is complex, and in most cases, the electron consumption associated with chain transfer or termination is typically neglected-unless the focus is specifically on termination pathways or chain extension efficiency⁴. Therefore, to estimate the electron demand, we assume that each polymer chain requires only one electron for initiation, corresponding to a single electron transfer to CTA to generate an initiating radical. Radical loss during chain propagation is thus disregarded. Under this assumption, the theoretical number of electrons required for polymerization ($n_{e,polymerization}$) can be calculated using Eq. 4⁵:

$$n_{e,polymerization} = n_{chains} = [CTA] \times V \times N_A \quad (\text{Eq. 4})$$

where n_{chains} is the number of polymer chains formed, equivalent to the number of initiating radical events in the system; $[CTA]$ is the concentration of the chain transfer agent (CTA); V is the reaction volume; N_A is Avogadro's constant ($6.022 \times 10^{23} \text{ mol}^{-1}$).

To estimate the theoretical ratio of electrons from microbes used specifically for CTA reduction, polymerization was carried out using 1 M monomer, 2 mM CTA, and 5 μM riboflavin in M9 minimal medium supplemented with 20 mM lactate as the electron donor. After 6 hours of reaction, a monomer conversion exceeding 95% was achieved. Subsequently, the reaction solution was filtered, and HPLC analysis revealed that 4.7 mM lactate had been consumed during the reaction. Based on the concentrations of CTA and lactate, we calculated the theoretical number of electrons required to initiate polymerization and compared it to the total number of electrons that could be generated from complete oxidation of the consumed lactate (4 electron equivalents per molecule). The calculation indicated that approximately 10% of the electrons in microbes were utilized for CTA reduction and initiation of polymerization in our system.

4. As we know, some nanoparticles can help the EET or photoelectron uptake, can it be integrated to this system?

Response: We greatly appreciate the Reviewer for this insightful comment. We fully agree that nanoparticles have been shown to enhance extracellular electron transfer (EET) and photoelectron uptake, and their integration into microbial-electronic systems represents a promising direction. Indeed, several studies have reported the use of nanoparticles as photocatalysts in PET-RAFT polymerization^{6, 7, 8, 9}. In the context of our microorganism-activated RAFT polymerization, it is conceptually feasible to replace flavins with redox-active nanoparticles, allowing them to serve as alternative electron shuttles that mediate electron transfer from microbial metabolism to the chain

transfer agents (CTAs), thereby initiating the polymerization reaction.

In fact, we are currently pursuing a follow-up study that explores the integration of microbial-nanoparticle hybrid systems as heterogeneous catalysts for RAFT polymerization. Preliminary results from these ongoing experiments indicated that nanoparticles can significantly improve electron transfer rates and catalytic performance, potentially enhancing both the efficiency and tunability of the polymerization process. We appreciate the Reviewer's suggestion, which encourages and promotes our future researches.

5. Although the authors provide the riboflavin concentration in Fig. 4b, it is required to present evidence if the riboflavin biosynthesis genes *ribADEHC* and *oprF* were expressed by qPCR or SDS-PAGE or Western blot?

Response: We thank the Reviewer for the valuable suggestion. Yes, direct evidence of gene expression would indeed strengthen the conclusion. To address this issue, we performed new experiments, i.e., the quantitative real-time PCR (qPCR) to confirm the enhanced transcriptional expression of both *ribADEHC* and *oprF* in the engineered *S. oneidensis* strains P-RBS4. As shown in **Supplementary Fig. 22**, the overexpression of these genes (including *ribA*, *ribD*, *ribE*, *ribH*, *ribC*, and *oprF*) were all detected in RNA level. These results demonstrated the overexpression of riboflavin biosynthesis genes *ribADEHC* and *oprF*.

6. Please provide the *c*-Cyts gene knockout/deletion design details.

Response: Thanks! The detailed procedure of the *c*-Cyts gene knockout/deletion design was added in **Line 157-171** of the revised Supplementary Information.

7. The authors may provide more experiment data on the stability of BioEPC-RAFT polymerization system, for example, system size, pH, and temperature.

Response: We appreciate the Reviewer for this constructive suggestion. To evaluate the robustness and stability of the polymerization system, we systematically studied this microorganism-triggered RAFT process under varying conditions, including reaction pH, temperature, and system volume. The results are presented in **Supplementary Fig. 15** and are summarized as follows:

To evaluate the pH stability of the polymerization system, reactions were conducted at pH values of 6.0, 6.5, 7.0, 7.5, and 8.0. The highest monomer conversion was observed at pH 7.0, indicating this as the optimal condition. Deviations from this value, either towards more acidic or basic conditions, resulted in a noticeable decrease in conversion efficiency (**Supplementary Fig. 15a**). To assess the temperature stability, polymerization was performed at 25 °C, 30 °C, 35 °C, and 40 °C. The system showed optimal performance at 20°C-30°C, while higher temperatures led to a significant

decrease in polymerization activity (**Supplementary Fig. 15b**). To investigate system volume scalability, reactions were carried out in volumes ranging from 0.5 mL to 100 mL. The monomer conversion remained consistently high across all scales tested, indicating excellent volumetric robustness and potential for upscaling (**Supplementary Fig. 15c**). These results demonstrate the practical applicability and operational robustness of our polymerization system.

8. The authors proposed this controlled radical polymerization can be controlled by light ON/OFF, but how to maintain the activity or viability of *S. oneidensis* if they perform a long-term reaction?

Response: We appreciate the Reviewer for raising this important question. To address the concern regarding the maintenance of *S. oneidensis* activity during long-term reactions, we performed additional experiments by extending reaction time. These experiments were designed to evaluate both the polymerization performance and microbial viability under prolonged reaction time. As shown in **Supplementary Fig. 17**, the results demonstrated that the polymerization process remained well-controlled even under extended light cycling. After a 24-hour reaction period, the final monomer conversion reached 91%, confirming the robustness and temporal controllability of this RAFT polymerization system. To further assess the viability of *S. oneidensis* over the course of the long-term reaction, we performed colony-forming unit (CFU) assays. The results showed a gradual decline in viability: approximately 95% of the initial viability was retained after 6 hours of polymerization, and around 65% remained viable after 24 hours. These results indicated that the polymerization system maintained good biocompatibility, allowing *S. oneidensis* to sustain sufficient viability over extended timeframes to support continuous electron transfer and efficient catalytic polymerization.

These revised contexts were highlighted in **Line 331-341** of the revised manuscript and the **Supplementary Fig. 17**, respectively.

9. In the Methods, how did the authors sample since they need to keep the oxygen limited condition?

Response: We thank the Reviewer for highlighting this important experimental detail. To maintain strict oxygen-limited conditions during sampling, the entire reaction flask was carefully transferred into an anaerobic glove box. The glove box was continuously purged with high-purity nitrogen to ensure that the internal oxygen concentration remained below 0.1 ppm. All sampling procedures-including handling, transferring, and aliquoting of the reaction mixture-were performed entirely within the glove box to avoid any exposure to atmospheric oxygen.

These revised contexts were added to **Methods** in the revised Supplementary Information (**Line 185-186 and Line 197-198**).

Reviewer #2 (Remarks to the Author):

Song and co-workers report the development of a bio-electro-photocatalytic process to directly reduce the C-S bond of thiocarbonylthio RAFT agents to initiate controlled radical polymerisations. Their approach combines electrocatalytic bacterium and photoactive components such as riboflavin and flavin mononucleotide (FMN). They demonstrate that their RAFT process retains control over molar mass and MW distribution, and that this process can be temporally controlled depending on irradiation and concentrations of the reagents.

While this approach is certainly novel, I do not believe this is a significant step forward for either living cell mediated RAFT polymerisation, which has been demonstrated via a few different approaches as described in the introduction. Neither is this a huge leap forward for photocatalytic RAFT polymerisation which has been extensively studied. The merger of these is interesting, but does not warrant publication in such a prestigious journal. I think that this could be resubmitted elsewhere as the findings are sound, figures are presented well and the paper is written very well.

Response: We appreciate the Reviewer for these insightful comments. We wish to take this opportunity to articulate the novelty of our study and how our work goes beyond the existing strategies and studies in the field of living cell-triggered RAFT polymerization.

Firstly, the **Introduction** of our manuscript seems not able to clearly define the concept of living cell-mediated RAFT polymerization, which led to misunderstanding. The living cell-mediated RAFT polymerization can generally be classified into two types: (i) RAFT polymerization occurred in cells, while the components of cells do not participate in the free radical polymerization processes, such as radical generation, chain propagation and termination. In these cases, the cells essentially provide a place for the polymerization to occur, which is typically driven by external stimuli (e.g., light)^{10, 11, 12, 13}, and the cells does not affect the polymerization processes (e.g., polymerization rate or initiation efficiency); (ii) Living cell-triggered RAFT polymerization, where the initiation of polymerization is driven by the metabolic activity of the cells¹⁴. In this case, the metabolic state of the cell is essential for radical generation, which enables regulation of the polymerization processes.

In the **Introduction**, we discussed several examples on the first type of living cell-mediated RAFT polymerization, where the cells have no impact or regulation on the polymerization processes. For instance, Hawker et al. developed a cyto-compatible PET-RAFT platform using Eosin Y as a photocatalyst to directly engineer the surfaces of live yeast and mammalian cells via cell surface-initiated polymerization¹³. This enabled active modulation of cellular phenotypes and cell-cell interactions. Geng et al. further extended this concept to intracellular PET-RAFT polymerization, developing a photoactivatable prodrug system for targeted cancer therapy¹². The resulting intracellular polymers were shown to induce cell cycle arrest, apoptosis, and necroptosis in cancer cells, effectively inhibiting tumor growth. Huang et al.

demonstrated that tyrosine residues on proteins can mediate RAFT polymerization under 405 nm light, achieving either extracellular polymerization on the yeast cell surface-enabling agglutination, or anti-agglutination, or intracellular polymerization within the yeast cells¹¹. In these studies, polymerization is triggered by external stimuli (e.g., light), and can proceed in the absence of cells. As such, the metabolism of the cells does not influence or regulate the polymerization processes, and the cells serve primarily as the space for the occurrence of polymerization reactions.

On the other hand, living cell-triggered RAFT polymerization, rewiring cellular metabolism to control the polymerization processes, is an important step toward sustainable, self-regenerable, and adaptable living materials with functional diversity of the synthetic polymers. However, since RAFT polymerization inherently requires a continuous supply of radicals to initiate and sustain the polymerization reaction¹⁵, this poses a significant challenge for the development of living cell activated RAFT polymerization. Up to now, only **one** study to achieve living cell-triggered RAFT polymerization has been reported by the Qiao group¹⁴, which relies on microbial reduction of exogenous initiators (e.g., diazonium salts) to produce aryl radicals that initiate polymerization. However, the use of exogenous radical initiators led to end-group heterogeneity and chain termination, which was an inherent issue of using free radical initiators in RAFT polymerization.

To overcome these limitations, we here developed a new strategy in which microbial cells directly reduce CTA to initiate RAFT polymerization, thereby avoiding the use of exogenous initiators. In our system, *Shewanella oneidensis*-secreted flavins (including riboflavin and flavin mononucleotide (FMN)) act as electron mediators, enabling the direct transfer of metabolic electrons from cells to the CTAs, which led to cleavage of the C-S bond of CTAs and generation of radicals to initiate RAFT polymerization (**Fig. 1**). In the extracellular electron transfer (EET) process, since the reduction potential of FL_{hq} is insufficient to reduce CTA, we thus introduced photoactivation to excite FL_{hq}, which reduced CTA via the photoinduced electron transfer (PET) process (**Fig. 2**). The photoactivation step helped overcome the thermodynamic barrier that prevented the direct electron transfer from the microbial cell to CTA.

To validate the essential role of cells, we carried out a series of control experiments. In the absence of cells, irradiated oxidized RF could not initiate polymerization (**Supplementary Table 11, entry 3**). When FL_{hq} was supplied alone, the monomer conversion was very low, indicating that FL_{hq} was consumed in the course of PET and could not be regenerated to sustain polymerization (**Supplementary Table 11, entry 5**). Additionally, when cells were killed during the polymerization process, the reaction stopped immediately. Moreover, we showed that modulating the carbon source (lactate) allowed us to regulate cellular metabolic activity, which in turn controlled the polymerization rate (**Supplementary Fig. 14**). Furthermore, knocking out genes encoding *c*-type cytochromes, which controlled the extracellular electron transfer (EET) efficiency, significantly reduced the polymerization rate (**Fig. 4d**). These experiments provided strong evidence that our polymerization system required active cellular metabolism, and that light was employed to overcome the thermodynamic barrier of

electron transfer, rather than functioning as a direct photo-initiator. Thus, our polymerization system should be categorized as living cell-activated RAFT polymerization, enabled by photo-assisted microbial electron transfer. By enabling direct microbial reduction of CTAs via extracellular electron transfer (EET), our system avoided the use of problematic exogenous initiators that led to end-group heterogeneity and subsequently impacted the quality of the synthesized polymers. Thus, our study was a meaningful advancement over previous approaches. We believe that our work constitutes a significant step forward in the field of living cell-triggered RAFT polymerization.

Secondly, as the Reviewer suggested that photocatalytic RAFT polymerization has indeed been extensively studied in recent years, which has enabled synthesis of polymers with structural complexities and functionalities for advanced applications⁸. However, we should emphasize that our system is fundamentally different from these photocatalytic RAFT polymerizations: our system is not a traditional photoinitiated polymerization system, but a living cell-triggered RAFT polymerization platform, in which photoactivation serves only as a step in assisting microbial electron transfer to enable the activation of chain transfer agents to generate radicals for initiation of the polymerization. Our approach successfully addresses the major unresolved issues in biologically driven RAFT polymerization, and we believe it represents a significant conceptual and technical innovation in living cell-triggered RAFT polymerization.

For this to be considered in a higher impact journal I would want to see an application which exploits the merger of the bio and photocatalytic RAFT process.

Response: We really appreciated the Reviewer for this valuable suggestion. We fully agree that demonstrating an application leveraging the synergy between the biological and photocatalytic components of our RAFT system would enhance the impact of our study. As a proof of concept, we thus performed new experiments to explore our microorganism-triggered RAFT system for high-throughput polymer synthesis.

Due to complex physiological activities of microbial cells and their responses to external stimuli, the rational design of functional materials via microorganism-triggered RAFT polymerization remains a significant challenge. High-throughput polymer synthesis and screening has emerged as an attractive alternative strategy, enabling the rapid identification of structure-property relationships and the discovery of novel materials⁸. However, one of the key challenges in high-throughput polymerization was to conduct efficient reactions in small volumes and under ambient conditions. Our *S. oneidensis*-triggered RAFT polymerization system had the feature of oxygen tolerance and high polymerization efficiency, which were explored to facilitate high-throughput polymer synthesis.

First, to assess whether RAFT polymerization could proceed efficiently without deoxygenation in small-volume systems, we evaluated the polymerization kinetics of DMA in 96- and 384-well microtiter plates, with reaction volumes of 200 μ L and 50

μL , respectively. Under blue light illumination, all polymerizations reached high conversion ratios with low dispersity (**Supplementary Table 17**, entry 1 and 2). The polymerization kinetics exhibited a linear correlation of $\ln([M_0]/[M])$ versus reaction time, indicating the livingness of the well plate polymerization (**Supplementary Fig. 30a** and **30b**). Notably, there was no significant difference in polymerization rates between the 50 μL and 200 μL polymerization reaction systems (**Supplementary Fig. 30a**), demonstrating the robustness of the system across varying scales.

Second, to demonstrate the applicability of *S. oneidensis*-triggered RAFT polymerization for synthesizing polymers with a range of molecular weights, we constructed a PDMA library targeting degrees of polymerization (DPs) from 100 to 2000 in a 96-well format. As shown in **Supplementary Table 17**, all polymerizations achieved high monomer conversions (**Supplementary Table 17**, entry 3-7). The molecular weight dispersity showed a slight increase with the growth of polymer molecular weight, which was maintained at a well-controlled fashion even at high DPs ($\mathcal{D} < 1.27$). GPC characterization showed a monomodal peak with a small tail, confirming that well-defined polymers with a wide range of molecular weights can be achieved (**Supplementary Fig. 30c**). Together, these results indicated that the *S. oneidensis*-triggered polymerization system provided a robust and efficient platform for high-throughput RAFT polymerization, offering a promising approach for the rapid synthesis and screening of polymer libraries.

These results were put in **Line 519-551** of the revised manuscript and **Supplementary Fig. 30** and **Supplementary Table 17**.

Finally, i also would recommend the authors to carefully explain how this new RAFT process can be distinguished from traditional iniferter RAFT polymerisation, as that is also a photomediated process but does not need bacteria or associated photo-catalytic molecules hence one could argue that the presented bio-photocatalytic approach is over complex.

Response: We sincerely thank the Reviewer for this insightful comment. We agree that it is important to clearly differentiate our system from traditional iniferter RAFT polymerization. This can be analyzed from two key perspectives: the reaction mechanism and the composition of the polymerization system:

(1) Traditional iniferter RAFT polymerization (as shown in **Fig. R1**): In iniferter RAFT polymerization¹⁶, CTAs (thiocarbonylthio compounds) act as an *initiator-transfer agent-terminator*. Upon light irradiation, CTAs undergo homolytic cleavage of the C-S bond, generating a carbon-centered radical (to initiate polymerization) and a stable sulfur-centered radical (to mediate reversible termination and chain transfer). This process requires only light to activate CTAs, without the need for any exogenous photoinitiators or photocatalysts. Thus, it is categorized as a photo-initiated and photo-regulated RAFT process, which is simple in design and execution.

Figure R1. The proposed mechanism of photoiniferter polymerization. CTAs directly absorb light to become an excited state, which is followed by the homolysis of the C-S bond to produce an active initiating carbon-centered radical and a persistent thiocarbonylthio radical. The initiating radicals can participate in degenerative chain transfer, and the TCT radical can deactivate active radical species in a reversible fashion.

(2) Living cell-activated RAFT polymerization (as shown in **Fig. R2**): By contrast, our system is a living cell-activated RAFT polymerization system, which is fundamentally different from the photoiniferter polymerization. In our system, the microbial extracellular electron transfer (EET) machinery is directly responsible for reducing CTA to initiate polymerization. Light is introduced only to photoexcite the reduced flavin species (FL_{hq}) in order to overcome the unfavorable thermodynamics of electron transfer from flavins to CTA. Our experiments clearly demonstrated that light alone was insufficient to initiate polymerization in the absence of microbial cells, highlighting that the polymerization was driven by the *S. oneidensis* cells with the assistant of light illumination. Therefore, our system cannot be considered a photoinitiated RAFT process.

Figure R2. The proposed mechanism of living cell (*S. oneidensis*)-triggered RAFT polymerization. To eliminate the impact of initiators, the microbial cells were used to directly reduce CTA to initiate RAFT polymerization. Oxidized flavins (FL) as the electron mediators were reduced to the fully reduced form of flavins hydroquinone (FL_{hq}) by the *S. oneidensis* cells. Then, under photoirradiation, FL_{hq} can reduce CTA via photoinduced electron transfer (PET) to form radicals that subsequently initiated the polymerization. The resulting oxidized flavins (FL) is then reduced back to FL_{hq} by the *S. oneidensis* cells, thereby completing the biocatalytic cycle.

It is true that iniferter polymerization does not require a photocatalyst or photoinitiator, while relies on direct photoactivation of CTA. However, the goal and mechanism of our system are fundamentally different: we aimed to develop a biologically driven RAFT platform, where the presence and metabolic activity of live cells are essential to the polymerization process. In addition, the iniferter polymerization systems generally require relatively high-intensity light, and the corresponding CTA structures are often sensitive to photodegradation, leading to limited photostability and potential side reactions⁸. In contrast, our system operates under low-intensity blue light, and CTAs we used exhibit excellent photostability under these conditions, resulting in a more stable and controllable photo environment.

In summary, our polymerization system is mechanistically and functionally distinct from the traditional iniferter RAFT polymerization systems. Our system represents a meaningful advancement in the field of living cell-triggered polymerization, enabling a new mode of integrating microbial electron flow with controlled radical polymerization.

Therefore, I do not recommend this to be published in nature communications and the authors consider more field-specific journals.

Response: We sincerely appreciated the Reviewer for the careful evaluation and constructive feedback, which promotes us to pondering more deeply on the scientific novelty of our study, and how it fits the interdisciplinary scope of *Nature Communications*.

Our study integrates elements from multiple disciplines (including synthetic biology, bioelectrochemistry, photochemistry, and polymer chemistry) to create a new approach and platform to enable living cell-triggered polymerization reactions. Rather than being an incremental development within one field (e.g., polymer chemistry or microbial catalysis), we proposed a conceptually new mechanism by coupling microbial metabolism and light-assisted redox catalysis to directly drive RAFT polymerization without exogenous initiators.

We believe this cross-disciplinary approach is well suited to the broad and high-impact platform of *Nature Communications*, especially as the development of programmable, biologically integrated materials is increasingly recognized as a key frontier in both material science and synthetic biology. Our work not only offers new insights into interface engineering of the microbe and polymerization systems, but also lays a groundwork for development of living material systems with bio-responsive and tunable control properties.

Reviewer #3 (Remarks to the Author):

This manuscript reports an *S. oneidensis*-activated photo-RAFT polymerization process. It utilizes genetically engineered *S. oneidensis* to synthesize and secrete flavins (riboflavin and FMN) as photo-initiators. The riboflavin-photoactivated RAFT process is known. The novel aspect of this work is that the photocatalyst is produced via a bioprocess.

Response: We sincerely appreciate the Reviewer for recognition of the novelty in our approach, particularly the use of genetically engineered *Shewanella oneidensis* to biosynthesize and secrete flavins as mediators for initiating RAFT polymerization upon illumination. While the riboflavin-photoactivated RAFT polymerization has been previously reported (e.g., Boyer *et al.*, *Polym. Chem.*, 2019, **10**, 4643–4654)¹⁷, our polymerization system is mechanistically and conceptually distinct from this prior study. We are grateful for the opportunity to elaborate on these differences and to further clarify the novel aspects of our microorganism-triggered RAFT strategy, as shown below:

(1) Previously reported riboflavin-mediated RAFT polymerization (**Fig. R3**): In the study reported by the Boyer's group¹⁷, the authors developed a two-component, vitamin-inspired, and thus biocompatible RAFT polymerization system consisting of a bioactive form of riboflavin (FMN) and vitamin C (ascorbic acid) as the photoinitiation pair, under blue LED irradiation. In this system, the photoexcited FMN is reduced by vitamin C (ascorbic acid) to the reduced FMN (FMN_{red}) via electron transfer. The resulting FMN_{red} subsequently reduces molecular oxygen to generate hydroxyl radicals, which act as initiators to trigger the free radical polymerization. The overall reaction mechanism resembles conventional RAFT polymerization, in which the initiation relies on exogenously generated radicals, while the chain transfer agent (CTA) functions solely to regulate the polymerization process via reversible chain transfer. However, the use of external radical initiators inevitably leads to end-group heterogeneity in the resulting polymers^{18, 19}. Additionally, the system requires the oxidized FMN as the photocatalyst and depends on reductive quenching to generate the catalytically active reduced form of FMN (FMN_{red}).

Figure R3. The proposed mechanism of photoinitiation by FMN in conjunction with vitamin C (ascorbic acid, AH-). Under blue light, the excited state FMN is photo-reduced by exogenous

vitamin C. Oxidation of the photo-reduced species generates superoxide and hydrogen peroxide by one or two electron transfer processes. Hydrogen peroxide can also be formed by disproportionation of superoxide. Finally, initiating hydroxyl radicals can be subsequently generated by direct reduction of hydrogen peroxide by excess vitamin C.

(2) Our microorganism-activated RAFT system (**Fig. 1 and Fig. R2**): Our study demonstrates a microbe-driven RAFT polymerization system where the metabolic activity of *Shewanella oneidensis* directly reduces the chain transfer agent (CTA), initiating radical formation and subsequent polymerization. To address the issue of end-group heterogeneity caused by microbial reduction of exogenous radical initiators, we employed genetically engineered *S. oneidensis* to biosynthesize and secrete flavins, which function as the electron mediators. These shuttles (flavins) intracellularly generated electrons, which were transferred to CTA to induce homolytic cleavage of the C-S bond in CTA and generate alkyl radicals to initiate polymerization. This strategy eliminated the need for externally added radical initiators or catalysts. However, the redox potential of flavin hydroquinone (FL_{hq}, i.e., the reduced form of flavins that are produced via microbial electron transfer) is not sufficiently negative to reduce CTAs under physiological conditions, thus posing a thermodynamic limitation for the electron transfer process. To overcome this challenge, we introduced a photoactivation strategy, which activated FL_{hq} under blue light to its excited state (FL_{hq}*), enabling the reduction of CTA via the photoinduced electron transfer (PET) mechanism.

As illustrated in **Figure 1**, the proposed mechanism involves three key steps: (1) the oxidized flavins (FL) function as redox mediators and are reduced to FL_{hq} via microbial extracellular electron transfer (EET) fueled by lactate metabolism of *Shewanella oneidensis*. (2) FL_{hq} are photoexcited to FL_{hq}*, which transfer electrons to CTA to generate alkyl radicals that initiate RAFT polymerization, then return to the ground-state oxidized form of flavins (FL). (3) FL are then continuously reduced by accepting new electrons from the microbial electron transport chain to form FL_{hq}, completing the catalytic cycle via the microbial electron transfer process. In this integrated system, the chain transfer agent (CTAs) serves also as the radical source, allowing precise polymer growth, while fully avoiding the use of external radical initiators and the associated end-group heterogeneity.

There are two major questions here:

Do you need the cells after the FL_{hq} is formed? Can you kill the cells and still conduct the polymerization?

Response: We thank the Reviewer for this important and insightful question regarding the necessity of live cells after the formation of reduced flavin hydroquinone (FL_{hq}). To address this question, we performed a set of control experiments to evaluate whether the polymerization can proceed in the absence of living cells after FL_{hq} is generated.

In the first experiment, we prepared a cell-riboflavin (RF) mixture by combining *S. oneidensis* cells ($OD_{600} = 5$) with 20 μM RF in M9 medium. The solution was degassed by nitrogen purging for 10 minutes to remove dissolved oxygen and then incubated under sealed, anaerobic conditions for 10 minutes until the yellow RF solution turned colorless, indicating complete reduction of RF to RF_{hq} . The solution was then transferred into an anaerobic glovebox and filtered through a 0.22 μm membrane to remove all cells, yielding a cell-free RF_{hq} -containing solution. Meanwhile, the mixture of monomer and CTA, as well as M9 medium, were also purged with nitrogen to remove dissolved oxygen and subsequently transferred into the anaerobic glovebox. The polymerization mixture was then assembled using this RF_{hq} solution (final $[\text{RF}_{\text{hq}}] = 5 \mu\text{M}$, $[\text{DMA}] = 1 \text{ M}$, $[\text{CTA}] = 2 \text{ mM}$, $[\text{DMSO}] = 5\%$, total volume = 2 mL), and irradiated with blue LED light at room temperature. A parallel reaction in presence of live *S. oneidensis* cells ($OD_{600} = 1$) under identical conditions served as the control.

As shown in **Supplementary Table 11**, the cell-free system achieved only ~25% monomer conversion after 6 hours, whereas the system containing live cells reached ~95% conversion under the same condition. This suggested that while RF_{hq} alone can initiate polymerization to a certain extent, sustained polymerization and high conversion clearly depend on the continuous presence of viable cells.

To further determine whether live cells are required throughout the polymerization process, we conducted a time-staged cell-killing experiment. At different time points (0.5 h, 1 h, and 2 h) during the polymerization reaction, the reaction mixtures were heated at 80°C for 20 minutes to inactivate the cells. After heat treatment, polymerization was continued under light irradiation for the remaining time to ensure all reactions ran for a total of 6 hours. These results are summarized in **Supplementary Fig. 16**. In all cases, monomer conversion essentially stalled following cell inactivation, confirming that live cells are essential not only at the initiation stage but also throughout the entire polymerization process.

In conclusion, our results clearly demonstrated that the presence of live *S. oneidensis* cells is essential for effective microorganism-activated RAFT polymerization. The cells not only biosynthesize and secrete flavins as electron mediators, but also serve as a continuous intracellular electron source. These electrons are required to regenerate the reduced flavin species (FL_{hq}) from the oxidized flavin (FL), thereby sustaining the photocatalytic cycle.

These results were put in **Line 230-238** of the revised manuscript.

2. Is this process oxygen-tolerant? Why is it, or why is it not?

Response: We thank the Reviewer for raising this important question regarding the oxygen tolerance of our microorganism-activated RAFT polymerization system. Radical polymerization usually requires strictly anoxic conditions since the initial radicals could be quenched in the presence of oxygen²⁰. *S. oneidensis* is a facultative anaerobe and preferentially respire on oxygen, which has been reported to quickly

consume dissolved oxygen by microbial metabolism in ATRP²¹. Therefore, to verify the dissolved oxygen tolerance of our method without degassing the reaction mixtures to retain active polymerization, we measured the general feasibility of aerobic polymerizations. As shown in **Supplementary Fig. 19**, the Vial-1 (anaerobic condition) achieved increased monomer conversion ratio (37%) compared to the Vial-2 (32% in aerobic condition) within the initial reaction of 1 hour. The initial polymerization rate of the system without oxygen removal was slightly lower than that of the de-oxygenated system, due to dissolved oxygen in the mixture quenched radicals to inhibit polymerization. With the elimination of oxygen by the respiration of *S. oneidensis* cells in the polymerization media, the polymerization rate of Vial-2 reached a similar level to that of Vial-1, and both eventually achieved monomer conversion ratio of ~80% after 4 hours' reaction. These results suggested that our polymerization system had good oxygen tolerance due to fast consumption and elimination of dissolved oxygen by *S. oneidensis*.

This exploration was added in **Line 342-358** of the revised manuscript.

Therefore, I am not entirely sure if this is truly a microorganism-activated RAFT process or merely a microorganism-produced photo-initiator system. I think this distinction is a major issue that must be clarified before accepting the claim in this paper.

Response: We sincerely appreciate the Reviewer for raising this important point! We apologize for any ambiguity in our original description that may have led to confusion regarding the mechanism of our microorganism-triggered RAFT polymerization system and its distinction from the microorganism-produced photo-initiator system.

The concept of living microorganism-triggered RAFT polymerization was first introduced by the Qiao group¹⁴, in which microbial metabolism was used to reduce exogenous diazonium salts to generate aryl radicals that initiate RAFT polymerization. However, the use of external radical initiators inherently introduced end-group heterogeneity and uncontrolled chain termination in the resulting polymers. To overcome these limitations, our goal was to develop a new form of microorganism-triggered RAFT polymerization, where the microbial electron flow directly drives the reduction of chain transfer agents (CTAs) to initiate polymerization, thereby eliminating the need for exogenous radical initiators.

In our system, the genetically engineered *Shewanella oneidensis* synthesizes and secretes flavins (riboflavin and FMN), which serve as electron mediators. These flavins shuttle electrons derived from microbial lactate metabolism to CTAs, initiating polymerization through C-S bond homolysis of CTAs. However, the direct reduction of CTAs by the reduced flavin hydroquinone (FL_{hq}) is thermodynamically unfavorable due to a mismatch in their redox potentials. To overcome this barrier, we introduced a photoexcitation step, namely, upon photoexcitation, FL_{hq} was converted to its excited state (FL_{hq}*), which possessed a significantly more negative reduction potential and could effectively reduce CTAs via the photoinduced electron transfer (PET) mechanism.

This photoactivation step was essential to enable microbial electrons-delivered through flavins-to reach CTAs and initiate radical polymerization. Therefore, while photochemistry is employed as an enabling element, the ultimate electron source remained from the microbial cell, and the radical generation process depended critically on continuous lactate metabolism of *Shewanella oneidensis*.

To further validate the essential role of living cells, we performed several control experiments (**Supplementary Table 11**). First, we showed that oxidized flavins (RF or FMN) produced by *S. oneidensis* under anaerobic conditions cannot initiate polymerization upon light irradiation alone (conversion < 5%). Second, when using purified reduced FL_{hq} as the sole photocatalyst in a cell-free system, the conversion increased modestly to ~25%. In contrast, in the presence of viable cells under otherwise identical conditions, conversion reached ~95%. Third, when cells were heat-inactivated at different time points during the polymerization process, the reaction promptly ceased, despite all other components remaining unchanged (**Supplementary Fig. 16**). These results collectively demonstrated that the live microbial cells were indispensable-not only for synthesizing flavins but also for continuously supplying electrons to regenerate the catalytically active FL_{hq} and sustained the redox cycle. Without continuous electron donation from the microbial metabolism of lactate (the carbon source), the polymerization process cannot proceed effectively.

Based on these mechanistic insights and experimental data, we recognized that our system was best described as a microorganism-activated RAFT polymerization. In our polymerization system, the microbial cells play three essential roles: (1) they act as biosynthetic factories for producing flavins, which serve as redox-active electron mediators; (2) they function as living electrodes or electron donors, continuously supplying electrons derived from lactate metabolism to regenerate the reduced form of flavins (FL_{hq}), thereby sustaining the catalytic cycle; and (3) they serve as oxygen scavengers, consuming dissolved oxygen through aerobic respiration to create an anaerobic environment for radical polymerization. The sustained electron transfer from microbial metabolism to the CTAs is critical for maintaining ongoing radical generation and efficient polymerization.

Taken together, these mechanistic insights supported our designation of the system as a microorganism-triggered RAFT polymerization. This terminology more accurately reflected a biologically driven polymerization process, in which the cells acted as electron source and light enabled electron transfer to the CTAs by overcoming inherent thermodynamic limitations. To reflect this conceptual refinement, we have revised the manuscript to consistently adopt this terminology (microorganism-triggered RAFT polymerization) in the **title, abstract, introduction, and conclusion**, thereby clarifying the specific and indispensable role of the microbial cells in our system.

Can you regulate cell growth to control the polymerization process? What are the experimental data supporting this?

Response: We thank the Reviewer for this thoughtful and insightful question! Indeed, regulating cell growth - and more broadly, tuning cellular metabolic activity - is a promising strategy to control the polymerization process in the microorganism-triggered RAFT polymerization system.

To explore this, we first investigated the effect of initial cell density on polymerization performance. Optimization experiments revealed that increasing the cell density in the reaction mixture led to higher monomer conversion and accelerated polymerization (**Supplementary Fig. 12**), indicating that cell concentration played a crucial role in determining polymerization efficiency.

Second, we assessed the impact of carbon source availability on cell metabolism and polymerization performance. As shown in **Supplementary Fig. 14**, under carbon starvation (no lactate), the monomer conversion remained low. When lactate (capable of generating four electron equivalents per molecule lactate) was used as the sole carbon source, both polymerization rate and conversion were significantly higher. In contrast, pyruvate, which yields only two electron equivalents per molecule, resulted in intermediate conversion. These results demonstrated that electron output from microbial metabolism, modulated by carbon source availability, directly regulated the polymerization process.

Third, we used synthetic biology approaches to engineer *S. oneidensis* with enhanced flavin biosynthesis and secretion. We introduced the *ribADEHC* gene cluster from *B. subtilis* under the control of different inducible promoters (Ptet, Ptac, Pbad, and ParcA), which yielded varying levels of flavin production and correspondingly different polymerization efficiencies (**Fig. 4b-c**). We further improved flavin export by modulating the expression of the outer membrane pore protein OprF using ribosome binding site variants (RBS1-RBS4). A strain with a weaker RBS (RBS4) produced the highest extracellular flavin concentration (~21.7 μM) and achieved the highest monomer conversion (~44%), likely due to reduced metabolic burden and enhanced flavin transport efficiency (**Fig. 4b-c**). These results confirmed that genetic modulation of flavin biosynthesis and export pathways had a direct and tunable impact on the RAFT polymerization efficiency.

Additionally, prior work from our group demonstrated that cell cycle regulation via genetic engineering can modulate growth rate and morphology (specifically, promoting faster cell division and smaller cell size), which enhanced the extracellular electron transfer rate²². While this strategy was not applied directly in the current study, these findings suggested that genetic control of cell growth dynamics may also be leveraged to further enhance polymerization kinetics. We greatly appreciate the Reviewer's insightful suggestion. This represented an important direction, and we intended to systematically investigate it in our future efforts.

In summary, our results demonstrated that the polymerization process can be effectively regulated by modulating growth conditions, metabolic activity, and genetic circuitry of *S. oneidensis*. Through altering cell density, carbon source availability, and targeted engineering of flavin biosynthesis and export, we established a versatile and

programmable platform for tuning the kinetics and efficiency of living microorganism-triggered RAFT polymerization.

Thus, it is premature to accept the paper with its current claim at this stage.

Response: We appreciate the Reviewer's critical assessment and fully respect the concern regarding the strength of the claims in our original submission. We acknowledge that certain mechanistic aspects and terminology were not sufficiently clarified, which could lead to potential misunderstanding.

In response, we thus have thoroughly revised the manuscript to more precisely define the scope and novelty of our study. In particular, we have adjusted the terminology throughout the manuscript-from "bio-electro-photocatalytic RAFT polymerization" to "microorganism-triggered RAFT polymerization" to better reflect the polymerization system, in which the microbial metabolism provided the electron source and redox mediators, while light facilitated the energetically unfavorable electron transfer to the CTAs. We hope that these comprehensive revisions could address the Reviewer's concerns and provide a more rigorously supported interpretation of our findings.

References

1. Sleutels THJA, Darus L, Hamelers HVM, Buisman CJN. Effect of operational parameters on Coulombic efficiency in bioelectrochemical systems. *Bioresource Technology* **102**, 11172-11176 (2011).
2. Masters BR. Molecular Fluorescence: Principles and Applications, Second Edition. *Journal of Biomedical Optics* **18**, 039901 (2013).
3. Lovley DR. The microbe electric: conversion of organic matter to electricity. *Current Opinion in Biotechnology* **19**, 564-571 (2008).
4. Perrier S. 50th Anniversary Perspective: RAFT Polymerization—A User Guide. *Macromolecules* **50**, 7433-7447 (2017).
5. Buback M. Kinetics and Mechanism of RAFT Polymerizations. In: *RAFT Polymerization* (ed Graeme Moad ER) (2021).
6. McClelland KP, Clemons TD, Stupp SI, Weiss EA. Semiconductor Quantum Dots Are Efficient and Recyclable Photocatalysts for Aqueous PET-RAFT Polymerization. *ACS Macro Letters* **9**, 7-13 (2019).
7. Jiang J, Ye G, Wang Z, Lu Y, Chen J, Matyjaszewski K. Heteroatom-Doped Carbon Dots (CDs) as a Class of Metal-Free Photocatalysts for PET-RAFT Polymerization under Visible Light and Sunlight. *Angewandte Chemie International Edition* **57**, 12037-12042 (2018).
8. Lee Y, Boyer C, Kwon MS. Photocontrolled RAFT polymerization: past, present, and future. *Chemical Society Reviews* **52**, 3035-3097 (2023).
9. Yaemsunthorn K, Macyk W, Ortyl J. Semiconductor photocatalysts in photopolymerization processes: Mechanistic insights, recent advances, and future prospects. *Progress in Polymer Science* **158**, 101891 (2024).
10. Zhang A, Zhao S, Tyson J, Deisseroth K, Bao Z. Applications of synthetic polymers directed toward living cells. *Nature Synthesis* **3**, 943-957 (2024).
11. Zhu M, *et al.* Tyrosine residues initiated photopolymerization in living organisms. *Nature Communications* **14**, 3598 (2023).
12. Abdelrahim M, *et al.* Light-mediated intracellular polymerization. *Nature Protocols* **19**, 1984-2025 (2024).
13. Niu J, *et al.* Engineering live cell surfaces with functional polymers via cytocompatible controlled radical polymerization. *Nature Chemistry* **9**, 537-545 (2017).
14. Nothling MD, Cao H, McKenzie TG, Hocking DM, Strugnell RA, Qiao GG. Bacterial Redox Potential Powers Controlled Radical Polymerization. *Journal of the American Chemical Society* **143**, 286-293 (2021).
15. Truong NP, Jones GR, Bradford KGE, Konkolewicz D, Anastasaki A. A comparison of RAFT and ATRP methods for controlled radical polymerization.

- Nature Reviews Chemistry* **5**, 859-869 (2021).
16. Hartlieb M. Photo-Iniferter RAFT Polymerization. *Macromolecular Rapid Communications* **43**, 2100514 (2022).
 17. Zhang T, Yeow J, Boyer C. A cocktail of vitamins for aqueous RAFT polymerization in an open-to-air microtiter plate. *Polymer Chemistry* **10**, 4643-4654 (2019).
 18. Hartlieb M. An acidic route for radical polymerizations. *Nature Synthesis* **3**, 291-292 (2024).
 19. Matyjaszewski K. Discovery of the RAFT Process and Its Impact on Radical Polymerization. *Macromolecules* **53**, 495-497 (2020).
 20. Parkatzidis K, Wang HS, Truong NP, Anastasaki A. Recent Developments and Future Challenges in Controlled Radical Polymerization: A 2020 Update. *Chem* **6**, 1575-1588 (2020).
 21. Fan G, Graham AJ, Kolli J, Lynd NA, Keitz BK. Aerobic radical polymerization mediated by microbial metabolism. *Nature Chemistry* **12**, 638-646 (2020).
 22. Yu H, *et al.* Accelerating cell division of *Shewanella oneidensis* to promote extracellular electron transfer rate for efficient pollution treatment. *Chemical Engineering Journal* **493**, 152765 (2024).

Point-by-point response to the Reviewers' Comments:

Reviewer #1 (Remarks to the Author):

The authors performed additional experiments and provided supporting data to address the reviewers' comments. The current manuscript exhibited highly valuable information and will benefit the readers. Therefore, I think this manuscript can be accepted.

Response: Many thanks for approving our work!

Reviewer #2 (Remarks to the Author):

Song and co-workers have resubmitted this manuscript following an initial round of review. The manuscript has been modestly revised with a two key elements. Firstly they have clarified the main novelty of the manuscript which was unclear previously.

They state that this novelty arises from being the first living cell initiated RAFT polymerisation where the radicals and photomediator are generated directly from the bacterial cells - which I agree with that this is the first time I have seen a system like this.

Secondly they have conducted further experiments demonstrating low volume and open-to-air polymerisations in 96 well plates, potentially showing how this system could be utilised in other applications.

Response: We sincerely appreciate the Reviewer for these positive comments.

Despite these changes and assertion of novelty, I feel that the advancements here are relatively niche and the extra experiments do not fully justify this relatively complex system.

For instance it has already been shown that bacterial mediated polymerisations can be done under non-inert conditions using exogenous initiators, either azo-aryl based like from Qiao and co-workers, or using a Fenton polymerisation from Gurnani and Rawson - ACS Macro Lett. 2022, 11, 8, 954–960. Hence the main novelty that remains is the ability to generate the flavin molecule from the engineered bacteria themselves. If they had shown that this is important in an application - e.g. for materials synthesis or

processing, then I would understand.

I therefore recommend that this could be transferred to communications chemistry or more specialised journal as the impact of this work is not strong enough for broad readership.

Response: Many thanks for these insightful comments!

We wish to articulate how this study represents a conceptual advance in construction of a recombinant cell-mediated controlled radical polymerization system, which enabled achievement of polymer materials with end-group fidelity. Such end-group fidelity feature is crucial in determining many physical and chemical properties of the polymers¹⁻⁴, which was firstly accomplished in the cell-based system without the need of exogeneous addition of initiators.

Living cell-mediated reversible deactivation radical polymerization (RDRP), which leverages cellular metabolic pathways to both initiate polymerization and synthesize functional polymers, holds significant potential for the construction of engineered living materials with broad application prospects. We acknowledge the pioneering contribution from the Qiao group and the Gurnani & Rawson group, who accomplished the living microorganism-activated RAFT polymerization^{5, 6}. Qiao *et al.* utilized microbial reduction of diazonium salts to generate aryl radicals that initiate polymerization⁵. Rawson *et al.* employed the microbial reduction of Fe³⁺ in conjunction with glucose oxidase to produce hydrogen peroxide, which subsequently drives hydroxyl radical formation via a Fenton-type reaction to initiate polymerization⁶. These studies, which rely on microbial reduction of exogenous radical precursors to generate initiating radicals, laid a foundation for development of microbe-driven polymerization. However, radicals generated from the exogenous initiators in these systems resulted in end-group heterogeneity and chain termination, which was an inherent issue of using free radical initiators in RAFT polymerization^{2, 4, 7, 8}. To eliminate the impact of exogeneous initiators on the quality of the synthesized polymers, we thus developed the *S. oneidensis*-triggered RAFT polymerization system. The polymerization is initiated directly via the microbial reduction of the chain transfer agent (CTA), which serves as the source of radicals, thereby eliminating the need of exogeneous addition of radical initiators.

To systematically highlight the novelty and significance of our work, we give a detailed comparative analysis in the following three aspects: 1) the polymerization mechanism and polymer synthesis; 2) the complexity and practicality of the polymerization system; and 3) the potential applications at the interface of polymer synthesis and synthetic biology.

1. The polymerization mechanism and polymer synthesis

(a) Polymerization mechanism: RAFT polymerization employed chain-transfer agents (CTAs, i.e., the thiocarbonylthio compounds) to achieve an equilibrium between propagating and dormant species via the degenerative chain-transfer mechanism. The RAFT polymerization process requires an initiator to provide a continuous source of radicals². In conventional RAFT systems, the unavoidable and often excessive use of radical initiators, which decompose to generate initiating radicals, can lead to undesirable α -termini (via direct initiation by the exogenous radical) or ω -termini (via termination of the propagating chain by the exogenous radical)^{4,9}. These side reactions ultimately result in increased termination events and loss of end-group fidelity. Recent studies by Qiao, Gurnani and Rawson have employed microbial metabolism to generate initiating radicals, such as aryl and hydroxyl radicals^{5,6}. However, the underlying mechanism of these two systems still fundamentally relies on exogenous radical precursors, which is basically the external initiator-activated RAFT polymerization systems (i.e., conventional RAFT systems). As a result, the radicals generated in these cell-mediated RAFT systems inevitably produce side reactions, leading to end-group heterogeneity and chain termination, thereby compromising the controllability of the polymerization process.

Well-defined polymeric microstructures (i.e., controlled monomeric sequence, high end group fidelity, etc.) are becoming increasingly desirable for a range of high-tech applications, with a significant drive to develop new polymerization strategies that display fine control over these features⁹. In contrast to previous systems, the *Shewanella oneidensis*-triggered RAFT polymerization system developed in our study meets these demands. In our system, the genetically engineered *S. oneidensis* synthesizes and secretes flavins (riboflavin and FMN), which serve as electron mediators. These flavins shuttle electrons derived from microbial lactate metabolism to the chain transfer agents (CTAs), and directly reduce CTAs to generate radicals to initiate polymerization. The CTA thus simultaneously serves as both the radical source and the chain transfer agent, effectively eliminating the need for exogenous initiators and their associated side reactions. This mechanism fundamentally addresses the issues of chain structural uniformity and end-group fidelity, offering a more controllable and cleaner route to precision polymer synthesis. Since the direct reduction of CTAs by the reduced flavin hydroquinone (FL_{hq}) is thermodynamically unfavorable due to mismatch in their redox potentials. To overcome this barrier, we introduced a photoexcitation step, namely, upon photoexcitation, FL_{hq} was converted to its excited state (FL_{hq}*), which possessed a significantly more negative reduction potential and could effectively reduce CTAs via the photoinduced electron transfer (PET) mechanism. This approach uniquely

integrates the sustainability of metabolically driven systems with the spatiotemporal precision of light control, establishing a synergistic light excited, and living cell-enabled RAFT polymerization mechanism.

(b) Polymer synthesis: In initiator-driven RAFT polymerization systems, there is a well-known trade-off between initiator concentration and polymerization control^{4, 10}. While higher concentrations of initiator can accelerate the polymerization rate, they also increase the likelihood of chain termination and side reactions. Therefore, to achieve controlled polymerization, the initiator concentration must be carefully optimized, often at the expense of overall monomer conversion. For example, in the RAFT polymerization system initiated by microbial reduction of diazonium salts reported by Qiao et al., the monomer conversion reached approximately 80%, and required multiple-times' addition of the initiator due to its cytotoxicity⁵. In the bacteria-initiated RAFT polymerization based on a modified Fenton reaction developed by Gurnani and Rawson, the conversion was even lower, around 60%⁶. In contrast, our polymerization system consistently achieves near-quantitative conversion (99%) across a broad range of monomers. This improved performance is attributed to the absence of exogenous radical initiators and the significantly reduced frequency of termination events.

(c) Block copolymer: Moreover, termination caused by initiators is further exemplified in the synthesis of block copolymers as a new aliquot of radical initiator is introduced together with the next monomer addition, thus leading to the gradual accumulation of terminated chains and resulting in higher dispersity and impure block copolymers^{7, 10}. In the bacteria-induced Fenton-RAFT polymerization developed by Gurnani and Rawson, chain extension experiments showed lower monomer conversion (monomer conversion ratio 49%) and broader molecular weight distributions ($D = 1.87$), suggesting a poor blocking efficiency⁶. The GPC trace revealed a bimodal distribution, indicating the coexistence of terminated macro-CTAs and newly initiated chains. In the bacteria-facilitated polymerization reported by Qiao et al., the high cytotoxicity associated with the diazonium-based initiator at elevated concentrations precluded the demonstration of block copolymer synthesis⁵. In contrast, our system successfully enables the synthesis of block copolymers (monomer conversion ratio 95%, $D = 1.15$), demonstrating excellent end-group fidelity and confirming the controlled/living character of the polymerization. This high level of end-group fidelity facilitates precise structural control over the resulting polymers, thereby laying a solid foundation for subsequent polymer functionalization, modification, and applications.

2. Polymerization system:

Compared to the currently reported microbe-mediated RAFT polymerization systems^{5, 6}, our system is not complex, offering significant advantages in terms of simplicity, operational ease, and practical implementation. Our microorganism-triggered RAFT polymerization system requires only culturing engineered microbial cells in standard M9 media, followed by the addition of monomers and CTA. Polymerization proceeds efficiently under low-power blue LED illumination without the need for deoxygenation, offering a mild, user-friendly, and scalable platform suitable for high-throughput or continuous operation.

In the Qiao's polymerization system⁵, in addition to cells, monomers, and CTA, a precise amount of diazonium salt initiator must be added. Due to the significant cytotoxicity of this initiator, it needs to be added in multiple portions throughout the polymerization process, resulting in a complex reaction setup that may complicate the operational procedure and scalability of continuous polymerization systems. In the Gurnani's system⁶, in addition to cells, monomers, and CTA, the polymerization system requires addition and fine-tuned regulation of multiple components, including the concentration of Fe²⁺ ions, glucose oxidase, and glucose, which involves relatively complex components and stringent control conditions.

3. Potential applications at the interface of polymer synthesis and synthetic biology

In this study, we employed synthetic biology to engineer microbial cells capable of endogenously synthesizing flavin-based electron mediators (including riboflavin and FMN), which can directly reduce the chain transfer agents (CTAs) to initiate polymerization, thereby eliminating the need for exogenous radical initiators. This approach represents the first demonstration of synergistic integration of synthetic biology and RAFT polymerization at the molecular mechanistic level, which constitutes one of the key innovations of our study, with significant scientific and technological implications.

On the one hand, by leveraging synthetic biology to reprogram microbial metabolism for *in situ* production of electron mediators, our system avoids the use of externally supplied electron donors or radical initiators, which greatly simplifies the reaction setup, enhances safety and controllability, and improves the overall sustainability of the process, which is a crucial factor for large-scale implementation of the RAFT polymerization process. On the other hand, our study demonstrates a practical example in applying synthetic biology to regulate controlled radical polymerization, thereby expanding the boundaries of synthetic biology in the field of functional materials

biosynthesis and contributing to the emergence of the interdisciplinary field of materials synthetic biology ¹¹. Moreover, as a forward-looking study, our work lays a critical mechanistic foundation for building a broader platform of engineering living materials synthesis in our future studies.

Building on the inherent scalability of synthetic biology, our research group has been actively expanding its applications beyond traditional biosynthetic pathways, particularly in the conversion of biomass into electrical energy ¹². Cellulosic biomass, as the most abundant renewable carbon resource globally, is critical for replacing fossil resources in the production of high-value chemicals and sustainable energy. Utilizing synthetic biology strategies, we have successfully engineered *Shewanella* strains by constructing a glucose and xylose co-utilization pathway, enabling the efficient conversion of cellulosic biomass into clean electricity. Leveraging this foundation, we are developing versatile living-cell platforms capable of synthesizing key catalytic mediators and driving controlled radical polymerizations, thereby opening new avenues in the field of materials synthetic biology. A few of our follow-up studies in leveraging synthetic biology (engineering microbial cells) and controlled free radical polymerization are shown below:

- (1) For example, we are developing a novel “one-pot” microbial catalytic system to directly facilitate anaerobic *in situ* ATRP polymerization. We recently constructed a complete heme biosynthetic pathway in the engineered microbes, enabling efficient heme biosynthesis. By coupling this pathway with extracellular electron transfer (EET) mechanisms to facilitate heme regeneration, we established a microbe-activated ATRP polymerization system. This approach eliminates the need for exogenous toxic metal catalysts that traditionally hinder biological compatibility, thereby offering a more biocompatible and sustainable strategy for controlled radical polymerizations.
- (2) Meanwhile, we recently also leveraged synthetic biology to expand the diversity of electron mediators and catalysts, thereby broadening the scope of bio-enabled polymerization systems. For example, we engineered the biosynthetic pathway of *S. oneidensis* for synthesis the photosensitizer indigo, achieving enhanced microbial production. The biologically synthesized indigo could be applied as a photocatalyst in the RAFT polymerization systems, further demonstrating the versatility of our microbial platform in driving controlled radical polymerizations.
- (3) We are also actively exploring the extension of synthetic biology from the biosynthesis of natural polymers to the controllable synthesis of non-natural functional polymers. Specifically, synthetic biology has allowed the synthesis of chemicals that can serve as monomers for downstream polymerization^{13, 14}.

Therefore, we are developing a biosynthetic pathway for the microbial production of acrylate monomers, enabling direct conversion of simple carbon sources (such as glucose or cellulosic biomass) into monomers (polymerizable building blocks). This pathway is further integrated with flavin biosynthesis, establishing a seamless progression from biomass to monomer, which subsequently leads to controlled free radical polymerization. Such fully biological route exemplifies the power of synthetic biology to construct sustainable polymer production systems. This advancement represents a significant step toward applying synthetic biology in materials (including functional polymers, biodegradable plastics, and biocompatible medical materials) biosynthesis.

Taken together, we believe our study offers not only an innovative mechanism that unites metabolic sustainability with precise spatiotemporal control, but also establishes a forward-looking research framework in the synergy of synthetic biology and polymer chemistry. We anticipate that this framework will provide both theoretical and experimental support for future developments in programmable functional material synthesis and the construction of living cell-based material platforms.

A few minor comments –

Fig 1. is a bit confusing, the end product seems to be in the top left which is where i would anticipate that the start of the 'flow' would be - so i suggest redrawing this

Response: Thank you for this helpful suggestion regarding **Figure 1**. We agree that the flow direction in the original design could be confusing. In the revised manuscript, we have redrawn **Figure 1** to better reflect the logical sequence of the process, starting from the monomers and progressing stepwise towards polymer synthesis. The end product is now positioned on the right-hand side of the figure to align with a natural left-to-right reading flow, which indeed improves clarity and readability.

Figure 1. Construction of the electroactive microorganism-triggered reversible addition-fragmentation chain-transfer (RAFT) polymerization system by photoexcited extracellular electron transfer of the genetically engineered *Shewanella oneidensis*.

Fig 3d - the trithiocarbonate groups are missing one sulfur atom in two of the polymer structures

Response: Thanks so much!! We have corrected the polymer structures in **Figure 3d** by adding the missing sulfur atom in the trithiocarbonate moieties. The updated figure has been included in the revised manuscript.

Figure 3d. Gel permeation chromatography (GPC) traces of block copolymer.

I would recommend that the data for SI table 17 is put in the manuscript to show the SEC curves and NMR spectra.

Response: Thank you for this valuable suggestion! We have moved the data originally provided in **Supplementary Table 17** into the main manuscript as a new figure (new **Figure 5**). This revision is of much help in improving the clarity and completeness of the manuscript.

Figure 5. GPC traces of high-throughput *S. oneidensis*-triggered RAFT polymerization. (a) Polymers PDMA (DP200) obtained from the corresponding 96 and 384-well plates. (b-f) The PDMA library targeting degrees of polymerization (DPs) from 100 to 2000 in a 96-well plate. (b)

DP100. (c) DP300. (d) DP500. (e) DP1000. (f) DP2000.

Reviewer #3 (Remarks to the Author):

The authors have taken my comments seriously and conducted a number of experiments to address my questions. I am convinced that the polymerization process can be enhanced by modulating cell growth, and that this is part of the mechanistic cycle. The paper is now in an acceptable form.

Response: Many thanks for approving our work!

References:

1. Lee, Y., Boyer, C. & Kwon, M.S. Photocontrolled RAFT polymerization: past, present, and future. *Chem Soc Rev* **52**, 3035-3097 (2023).
2. Truong, N.P., Jones, G.R., Bradford, K.G.E., Konkolewicz, D. & Anastasaki, A. A comparison of RAFT and ATRP methods for controlled radical polymerization. *Nature Reviews Chemistry* **5**, 859-869 (2021).
3. Antonopoulou, M.N., Truong, N.P. & Anastasaki, A. Enhanced synthesis of multiblock copolymers via acid-triggered RAFT polymerization. *Chem Sci* **15**, 5019-5026 (2024).
4. Perrier, S. 50th Anniversary Perspective: RAFT Polymerization—A User Guide. *Macromolecules* **50**, 7433-7447 (2017).
5. Nothling, M.D. *et al.* Bacterial Redox Potential Powers Controlled Radical Polymerization. *Journal of the American Chemical Society* **143**, 286-293 (2021).
6. Bennett, M.R. *et al.* Oxygen-Tolerant RAFT Polymerization Initiated by Living Bacteria. *ACS Macro Lett* **11**, 954-960 (2022).
7. Antonopoulou, M.-N. *et al.* Acid-triggered radical polymerization of vinyl monomers. *Nature Synthesis* **3**, 347-356 (2024).
8. Parkatzidis, K., Wang, H.S., Truong, N.P. & Anastasaki, A. Recent Developments and Future Challenges in Controlled Radical Polymerization: A 2020 Update. *Chem* **6**, 1575-1588 (2020).
9. McKenzie, T.G. *et al.* Beyond Traditional RAFT: Alternative Activation of Thiocarbonylthio Compounds for Controlled Polymerization. *Advanced Science* **3**, 1500394 (2016).
10. Zhou, Y.-N., Li, J.-J., Wang, T.-T., Wu, Y.-Y. & Luo, Z.-H. Precision polymer synthesis by controlled radical polymerization: Fusing the progress from polymer chemistry and reaction engineering. *Progress in Polymer Science* **130**, 101555 (2022).
11. Tang, T.-C. *et al.* Materials design by synthetic biology. *Nature Reviews Materials* **6**, 332-350 (2020).
12. Zhang, J. *et al.* Engineering *Shewanella oneidensis* to efficiently co-utilize glucose and xylose for converting cellulose hydrolysate from corn stover to electricity. *Chemical Engineering Journal* **505**, 159574 (2025).
13. Chae, T.U. *et al.* Biosynthesis of poly(ester amide)s in engineered *Escherichia coli*. *Nat Chem Biol* (2025).
14. Lee, N. *et al.* Retrobiosynthesis of unnatural lactams via reprogrammed

polyketide synthase. *Nature Catalysis* **8**, 389-402 (2025).